# MONST3R: A SIMPLE APPROACH FOR ESTIMATING GEOMETRY IN THE PRESENCE OF MOTION

**Junyi Zhang**[1]   **Charles Herrmann**[2,†]   **Junhwa Hur**[2]   **Varun Jampani**[3]   **Trevor Darrell**[1]
**Forrester Cole**[2]   **Deqing Sun**[2,*]   **Ming-Hsuan Yang**[2,4,*]

[1]UC Berkeley   [2]Google DeepMind   [3]Stability AI   [4]UC Merced

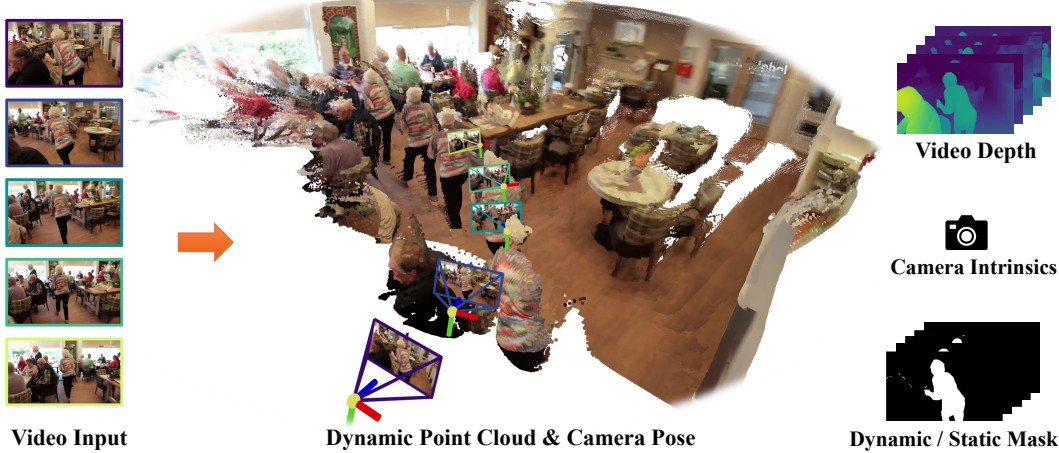

Figure 1: **MonST3R** processes a dynamic video to produce a time-varying dynamic point cloud, along with per-frame camera poses and intrinsics, in a predominantly feed-forward manner. This representation then enables the efficient computation of downstream tasks, such as video depth estimation and dynamic/static scene segmentation.

## ABSTRACT

Estimating geometry from dynamic scenes, where objects move and deform over time, remains a core challenge in computer vision. Current approaches often rely on multi-stage pipelines or global optimizations that decompose the problem into subtasks, like depth and flow, leading to complex systems prone to errors. In this paper, we present Motion DUSt3R (MonST3R), a novel geometry-first approach that directly estimates per-timestep geometry from dynamic scenes. Our key insight is that by simply estimating a pointmap for each timestep, we can effectively adapt DUSt3R's representation, previously only used for static scenes, to dynamic scenes. However, this approach presents a significant challenge: the scarcity of suitable training data, namely dynamic, posed videos with depth labels. Despite this, we show that by posing the problem as a fine-tuning task, identifying several suitable datasets, and strategically training the model on this limited data, we can surprisingly enable the model to handle dynamics, even without an explicit motion representation. Based on this, we introduce new optimizations for several downstream video-specific tasks and demonstrate strong performance on video depth and camera pose estimation, outperforming prior work in terms of robustness and efficiency. Moreover, MonST3R shows promising results for primarily feed-forward 4D reconstruction. Interactive 4D results, source code, and trained models are available at: https://monst3r-project.github.io/.

## 1 INTRODUCTION

Despite recent progress in 3D computer vision, estimating geometry from videos of dynamic scenes remains a fundamental challenge. Traditional methods decompose the problem into subproblems

---

†Project lead, *Equal contribution

such as depth, optical flow, or trajectory estimation, addressed with specialized techniques, and then combine them through global optimization or multi-stage algorithms for dynamic scene reconstruction (Luiten et al., 2020; Kumar et al., 2017; Bârsan et al., 2018; Mustafa et al., 2016). Even recent work often takes optimization-based approaches given intermediate estimates derived from monocular video (Lei et al., 2024; Chu et al., 2024; Wang et al., 2024b; Liu et al., 2024; Wang et al., 2024a). However, these multi-stage methods are usually slow, brittle, and prone to error at each step.

While highly desirable, end-to-end geometry learning from a dynamic video poses a significant challenge, requiring a suitable representation that can represent the complexities of camera motion, multiple object motion, and geometric deformations, along with annotated training datasets. While prior methods have centered on the combination of motion and geometry, motion is often difficult to directly supervise due to lack of annotated training data. Instead, we explore using *only* geometry to represent dynamic scenes, inspired by the recent work DUSt3R (Wang et al., 2024c).

For static scenes, DUSt3R introduces a new paradigm that directly regresses scene geometry. Given a pair of images, DUSt3R produces a pointmap representation - which associates every pixel in each image with an estimated 3D location (*i.e.*, $xyz$) and aligns these pair of pointmaps in the camera coordinate system of the first frame. For multiple frames, DUSt3R accumulates the pairwise estimates into a global point cloud and uses it to solve numerous standard 3D tasks such as single-frame depth, multi-frame depth, or camera intrinsics and extrinsics.

We leverage DUSt3R's pointmap representation to directly estimate geometry of dynamic scenes. Our key insight is that pointmaps can be estimated per timestep and that representing them in the same camera coordinate frame still makes conceptual sense for dynamic scenes. As shown in Fig. 1, an estimated pointmap for the dynamic scene appears as a point cloud where dynamic objects appear at multiple locations, according to how they move. Multi-frame alignment can be achieved by aligning pairs of pointmaps based on static scene elements. This setting is a generalization of DUSt3R to dynamic scenes and allows us to use the same network and original weights as a starting point.

One natural question is if DUSt3R can already and effectively handle video data with moving objects. However, as shown in Fig. 2, we identify two significant limitations stemming from the distribution of DUSt3R's training data. First, since its training data contains only static scenes, DUSt3R fails to correctly align pointmaps of scenes with moving objects; it often relies on moving foreground objects for alignment, resulting in incorrect alignment for static background elements. Second, since its training data consists mostly of buildings and backgrounds, DUSt3R sometimes fails to correctly estimate the geometry of foreground objects, regardless of their motion, and places them in the background. In principle, both problems originate from a domain mismatch between training and test time and can be solved by re-training the network.

However, this requirement for dynamic, posed data with depth presents a challenge, primarily due to its scarcity. Existing methods, such as COLMAP (Schönberger & Frahm, 2016), often struggle with complex camera trajectories or highly dynamic scenes, making it challenging to produce even pseudo ground truth data for training. To address this limitation, we identify several small-scale datasets that possess the necessary properties for our purposes.

Our main finding is that, surprisingly, we can successfully adapt DUSt3R to handle dynamic scenes by identifying suitable training strategies designed to maximally leverage this limited data and fine-tuning on them. We then introduce several new optimization methods for video-specific tasks using these pointmaps and demonstrate strong performance on video depth and camera pose estimation, as well as promising results for primarily feed-forward 4D reconstruction.

The contributions of this work are as follows:

- We introduce Motion DUSt3R (MonST3R), a geometry-first approach to dynamic scenes that directly estimates geometry in the form of pointmaps, even for moving scene elements. To this end, we identify several suitable datasets and show that, surprisingly, a small-scale fine-tuning achieves promising results for direct geometry estimation of dynamic scenes.
- MonST3R obtains promising results on several downstream tasks (video depth and camera pose estimation). In particular, MonST3R offers key advantages over prior work: enhanced robustness, particularly in challenging scenarios; increased speed compared to optimization-based methods; and competitive results with specialized techniques in video depth estimation, camera pose estimation and dense reconstruction.

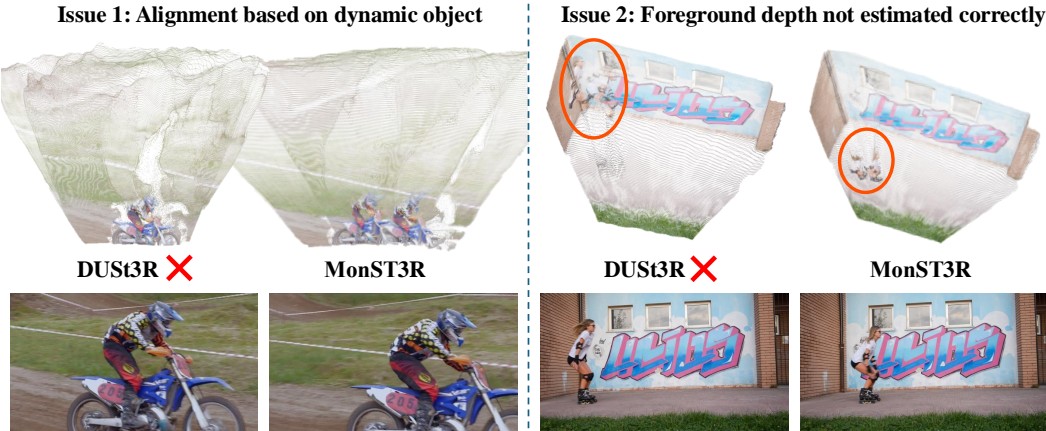

Figure 2: **Limitation of DUSt3R on dynamic scenes.** Left: DUSt3R aligns the moving foreground subject and misaligns the background points as it is only trained on static scenes. Right: DUSt3R fails to estimate the depth of a foreground subject, placing it in the background.

## 2 RELATED WORK

**Structure from motion and visual SLAM.** Given a set of 2D images, structure from motion (SfM) (Schönberger & Frahm, 2016; Teed & Deng, 2018; Tang & Tan, 2018) or visual SLAM (Teed & Deng, 2021; Mur-Artal et al., 2015; Mur-Artal & Tardós, 2017; Engel et al., 2014; Newcombe et al., 2011) estimate 3D structure of a scene while also localizing the camera. However, these methods struggle with dynamic scenes with moving objects, which violate the epipolar constraint.

To address this problem, recent approaches have explored joint estimation of depth, camera pose, and residual motion, optionally with motion segmentation to exploit the epipolar constraints on the stationary part. Self-supervised approaches (Gordon et al., 2019; Mahjourian et al., 2018; Godard et al., 2019; Yang et al., 2018) learn these tasks through self-supervised proxy tasks. Casual-SAM (Zhang et al., 2022) finetunes a depth network at test time with a joint estimation of camera pose and movement mask. Robust-CVD (Kopf et al., 2021) jointly optimizes depth and camera pose given optical flow and binary masks for dynamic objects. Our approach directly estimates 3D structure of a dynamic scene in the pointmap representation without time-consuming test-time finetuning.

**Representation for static 3D reconstruction.** Learning-based approaches reconstruct static 3D geometry of objects or scenes by learning strong 3D priors from training datasets. Commonly used output representations include point clouds (Guo et al., 2020; Lin et al., 2018), meshes (Gkioxari et al., 2019; Wang et al., 2018), voxel (Sitzmann et al., 2019; Choy et al., 2016; Tulsiani et al., 2017), implicit representation (Wang et al., 2021a; Peng et al., 2020; Chen & Zhang, 2019), *etc*.

DUSt3R (Wang et al., 2024c) introduces a pointmap representation for scene-level 3D reconstruction. Given two input images, the model outputs a 3D point of each pixel from both images in the camera coordinate system of the first frame. The model implicitly infers camera intrinsics, relative camera pose, and two-view geometry and thus can output an aligned points cloud with learned strong 3D priors. However, the method targets only static scenes. MonST3R shares the pointmap representation of DUSt3R but targets scenes with dynamic objects.

**Learning-based visual odometry.** Learning-based visual odometry replaces hand-designed parts of geometry-based methods (Mur-Artal et al., 2015; Mur-Artal & Tardós, 2017; Engel et al., 2017) and enables large-scale training for better generalization even with moving objects. Trajectory-based approaches (Chen et al., 2024; Zhao et al., 2022) estimate long-term trajectories along a video sequence, classify their dynamic and static motion, and then localize camera via bundle adjustment. Joint estimation approaches additionally infer moving object mask (Shen et al., 2023) or optical flow (Wang et al., 2021b) to be robust to moving objects while requiring their annotations during training. In contrast, our method directly outputs dynamic scene geometry via a pointmap representation and localizes camera afterwards.

**Monocular and video depth estimation.** Recent deep learning works (Ranftl et al., 2020; 2021; Saxena et al., 2024; Ke et al., 2024) target zero-shot performance and with large-scale training combined with synthetic datasets (Yang et al., 2024a;b) show strong generalization to diverse domains.

Table 1: **Training datasets** used fine-tuning on dynamic scenes. All datasets provide both camera pose and depth, and most of them include dynamic objects.

| Dataset | Domain | Scene type | # of frames | # of Scenes | Dynamics | Ratio |
|---|---|---|---|---|---|---|
| PointOdyssey (Zheng et al., 2023) | Synthetic | Indoors & Outdoors | 200k | 131 | Realistic | 50% |
| TartanAir (Wang et al., 2020) | Synthetic | Indoors & Outdoors | 1000k | 163 | None | 25% |
| Spring (Mehl et al., 2023) | Synthetic | Outdoors | 6k | 37 | Realistic | 5% |
| Waymo Perception (Sun et al., 2020) | Real | Driving | 160k | 798 | Driving | 20% |

However, for video, these approaches suffer from flickering (temporal inconsistency between nearby estimates) due to their process of only a single frame and invariant training objectives.

Early approaches to video depth estimation (Luo et al., 2020; Zhang et al., 2021) improve temporal consistency by fine-tuning depth models, and sometimes motion models, at test time for each input video. Self-supervised methods (Watson et al., 2021; Sun et al., 2023) are also explored to enhance temporal coherence without explicit annotations. Two recent approaches attempt to improve video depth estimation using generative priors. However, Chronodepth (Shao et al., 2024) still suffers from flickering due to its window-based inference, and DepthCrafter (Hu et al., 2024) produces scale-/shift-invariant depth, which is unsuitable for many 3D applications (Yin et al., 2021).

**4D reconstruction.** Concurrently approaches (Lei et al., 2024; Chu et al., 2024; Wang et al., 2024b; Liu et al., 2024) introduce 4D reconstruction methods of dynamic scenes. Given a monocular video and pre-computed estimates (*e.g*., 2D motion trajectory, depth, camera intrinsics and pose, *etc*.), the approaches reconstruct the input video in 4D space via test-time optimization of 3D Gaussians (Kerbl et al., 2023) with deformation fields, facilitating novel view synthesis in both space and time. Our method is orthogonal to the methods and estimate geometry from videos in a feed-forward manner. Our estimates could be used as initialization or intermediate signals for these methods.

## 3 METHOD

### 3.1 BACKGROUND AND BASELINES

**Model architecture.** Our architecture is based on DUSt3R (Wang et al., 2024c), a ViT-based architecture (Dosovitskiy et al., 2021) that is pre-trained on a cross-view completion task (Weinzaepfel et al., 2023) in a self-supervised manner. Two input images are first individually fed to a shared encoder. A following transformer-based decoder processes the input features with cross-attention. Then two separate heads at the end of the decoder output pointmaps of the first and second frames aligned in the coordinate of the first frame.

**Baseline with mask.** While DUSt3R is designed for static scenes as shown in Fig. 2, we analzye its applicability to dynamic scenes by using knowledge of dynamic elements (Chen et al., 2024; Zhao et al., 2022). Using ground truth moving masks, we adapt DUSt3R by masking out dynamic objects during inference at both the image and token levels, replacing dynamic regions with black pixels in the image and corresponding tokens with mask tokens. This approach, however, leads to degraded pose estimation performance (Sec. 4.3), likely because the black pixels and mask tokens are out-of-distribution with respect to training. This motivates us to address these issues in this work.

### 3.2 TRAINING FOR DYNAMICS

**Main idea.** While DUSt3R primarily focuses on static scenes, the proposed MonST3R can estimate the geometry of dynamic scenes over time. Figure. 1 shows a visual example consisting of a point cloud where dynamic objects appear at different locations, according to how they move.

Similar to DUSt3R, for a single image $\mathbf{I}^t$ at time $t$, MonST3R also predicts a pointmap $\mathbf{X}^t \in \mathbb{R}^{H \times W \times 3}$. For a pair of images, $\mathbf{I}^t$ and $\mathbf{I}^{t'}$, we adapt the notation used in the global optimization section of DUSt3R. The network predicts two corresponding pointmaps, $\mathbf{X}^{t;t \leftarrow t'}$ and $\mathbf{X}^{t';t \leftarrow t'}$, with confidence map, $\mathbf{C}^{t;t \leftarrow t'}$ and $\mathbf{C}^{t';t \leftarrow t'}$ The first element $t$ in the superscript indicates the frame that the pointmap corresponds to, and $t \leftarrow t'$ indicates that the network receives two frames at $t, t'$ and that the pointmaps are in the coordinate frame of the camera at $t$. The key difference from DUSt3R is that each pointmap in MonST3R relates to a single point in time.

**Training datasets.** A key challenge in modeling dynamic scenes as per-timestep pointmaps lies in the scarcity of suitable training data, which requires synchronized annotations of input images, camera poses, and depth. Acquiring accurate camera poses for real-world dynamic scenes is particularly challenging, often relying on sensor measurements or post-processing through structure from motion (SfM) (Schönberger et al., 2016; Schönberger & Frahm, 2016) while filtering out moving objects. Consequently, we leverage primarily synthetic datasets, where accurate camera poses and depth can be readily extracted during the rendering process.

For our dynamic fine-tuning, we identify four large video datasets: three synthetic datasets - PointOdyssey (Zheng et al., 2023), TartanAir (Wang et al., 2020), and Spring (Mehl et al., 2023), along with the real-world Waymo dataset (Sun et al., 2020), as shown in Tab. 1. These datasets contain diverse indoor/outdoor scenes, dynamic objects, camera motion, and labels for camera pose and depth. PointOdyssey and Spring are both synthetically rendered scenes with articulated, dynamic objects; TartanAir consists of synthetically rendered drone fly-throughs of different scenes without dynamic objects; and Waymo is a real-world driving dataset labeled with LiDAR.

During training, we sample the datasets asymmetrically to place extra weight on PointOdyssey (more dynamic, articulated objects) and less weight on TartanAir (good scene diversity but static) and Waymo (a highly specialized domain). Images are downsampled such that their largest dimension is 512.

**Training strategies.** Due to the relatively small size of this dataset mixture, we adopt several training techniques designed to maximize data efficiency. First, we only finetune the prediction head and decoder of the network while freezing the encoder. This strategy preserves the geometric knowledge in the CroCo (Weinzaepfel et al., 2022) features and should decrease the amount of data required for fine-tuning. Second, we create training pairs for each video by sampling two frames with temporal strides ranging from 1 to 9. The sampling probabilities increase linearly with the stride length, with the probability of selecting stride 9 being twice that of stride 1. This gives us a larger diversity of camera and scene motion and more heavily weighs larger motion. Third, we utilize a Field-of-View augmentation technique using center crops with various image scales. This encourages the model to generalize across different camera intrinsics, even though such variations are relatively infrequent in the training videos. We train the model with the same confidence-aware regression loss as DUSt3R.

### 3.3 DOWNSTREAM APPLICATIONS

**Instrinsics and relative pose estimation.** Since the intrinsic parameters are estimated based on the pointmap in its own camera frame $\mathbf{X}^{t;t\leftarrow t'}$, the assumptions and computation listed in DUSt3R are still valid, and we only need to solve for focal length $f^t$ to obtain the camera intrinsics $\mathbf{K}^t$.

To estimate relative pose $\mathbf{P}^* = [\mathbf{R}^*|\mathbf{T}^*]$, where $\mathbf{R}^*$ and $\mathbf{T}^*$ represent the camera's rotation and translation, respectively, dynamic objects violate the assumptions for methods relying on correspondences between *two views*, *e.g.*, epipolar matrix (Hartley & Zisserman, 2003) with 2D and Procrustes alignment (Luo & Hancock, 1999) with 3D correspondences. Instead, we leverage per-pixel 2D-3D correspondences within the *same view* and use PnP (Lepetit et al., 2009) to recover the relative pose:

$$\mathbf{R}^*, \mathbf{T}^* = \arg\min_{\mathbf{R},\mathbf{T}} \sum_{i \in \mathcal{I}} \left\| \mathbf{x}_i - \pi \left( \mathbf{K}^{t'} \left( \mathbf{R} \mathbf{X}_i^{t';t\leftarrow t'} + \mathbf{T} \right) \right) \right\|^2, \tag{1}$$

where $\mathbf{x}$ is the pixel coordinate matrix and $\pi(\cdot)$ is the projection operation $(x, y, z) \rightarrow (x/z, y/z)$. To improve the robustness to outliers, we use RANSAC (Fischler & Bolles, 1981) and define valid correspondences by taking a threshold of the estimated confidence mask, $\mathcal{I} = \{i \mid \mathbf{C}_i^{t';t\leftarrow t'} > \alpha\}$.

**Confident static regions.** We can infer static regions in frames $t, t'$ by comparing the estimated optical flow with the flow field that results from applying only camera motion from $t$ to $t'$ to the pointmap at $t$. The two flow fields should agree for pixels where the geometry has been correctly estimated and are static. Given a pair of frames $\mathbf{I}^t$ and $\mathbf{I}^{t'}$, we first compute two sets of pointmaps $\mathbf{X}^{t;t\leftarrow t'}, \mathbf{X}^{t';t\leftarrow t'}$ and $\mathbf{X}^{t;t'\leftarrow t}, \mathbf{X}^{t';t'\leftarrow t}$. We then use these pointmaps to solve for the camera intrinsics ($\mathbf{K}^t$ and $\mathbf{K}^{t'}$) for each frame and the relative camera pose from $t$ to $t'$, $\mathbf{P}^{t\rightarrow t'} = [\mathbf{R}^{t\rightarrow t'}|\mathbf{T}^{t\rightarrow t'}]$ as above. We then compute the optical flow field induced by camera motion, $\mathbf{F}_{\text{cam}}^{t\rightarrow t'}$, by backprojecting

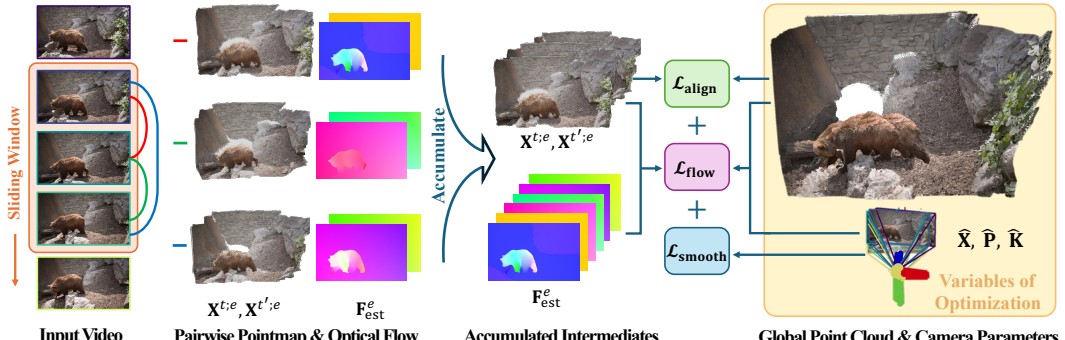

**Figure 3: Dynamic global point cloud and camera pose estimation.** Given a fixed sized of temporal window, we compute pairwise pointmap for each frame pair with MonST3R and optical flow from off-the-shelf method. These intermediates then serve as inputs to optimize a global point cloud and per-frame camera poses. Video depth can be directly derived from this unified representation.

each pixel in 3D, applying relative camera motion, and projecting back to image coordinate,

$$\mathbf{F}_{\mathrm{cam}}^{t \to t'} = \pi(\mathbf{D}^{t;t \leftarrow t'} \mathbf{K}^{t'} \mathbf{R}^{t \to t'} \mathbf{K}^{t-1} \hat{\mathbf{x}} + \mathbf{K}^{t'} \mathbf{T}^{t \to t'}) - \mathbf{x}, \tag{2}$$

where $\hat{\mathbf{x}}$ is the pixel coordinate matrix $\mathbf{x}$ in homogeneous coordinates, and $\mathbf{D}^{t;t \leftarrow t'}$ is estimated depth extracted from the point map $\mathbf{X}^{t;t \leftarrow t'}$. Then we compare it with optical flow (*i.e.*, $\mathbf{F}_{\mathrm{est}}^{t \to t'}$) computed by an off-the-shelf optical flow method (Wang et al., 2024d) and infer the static mask $\mathbf{S}^{t \to t'}$ via a simple thresholding:

$$\mathbf{S}^{t \to t'} = \left[\alpha > ||\mathbf{F}_{\mathrm{cam}}^{t \to t'} - \mathbf{F}_{\mathrm{est}}^{t \to t'}||_{\mathrm{L1}}\right], \tag{3}$$

with a threshold $\alpha$, $||\cdot||_{\mathrm{L1}}$ for smooth-L1 norm (Girshick, 2015), and $[\cdot]$ for the Iverson bracket. This confident, static mask is both a potential output and will be used in the later global pose optimization.

## 3.4 DYNAMIC GLOBAL POINT CLOUDS AND CAMERA POSE

Even a short video contain numerous frames (*e.g.* a 5-second video with 24 fps gives 120 frames) making it non-trivial to extract a single dynamic point cloud from pairwise pointmap estimates across the video. Here, we detail the steps to simultaneously solve for a global dynamic point cloud and camera poses by leveraging our pairwise model and the inherent temporal structure of video.

**Video graph.** For global alignment, DUSt3R constructs a connectivity graph from all pairwise frames, a process that is prohibitively expensive for video. Instead, as shown on the left of Fig. 3, we process video with a sliding temporal window, significantly reducing the amount of compute required. Specifically, given a video $\mathbf{V} = [\mathbf{I}^0, \ldots, \mathbf{I}^N]$, we compute pointmaps for all pairs $e = (t, t')$ within a temporal window of size $w$, $\mathbf{W}^t = \{(a, b) \mid a, b \in [t, \ldots, t+w], a \neq b\}$ and for all valid windows $\mathbf{W}$. To further improve the run time, we also apply strided sampling.

**Dynamic global point cloud and pose optimization.** The primary goal is to accumulate all pairwise pointmap predictions (*e.g.*, $\mathbf{X}^{t;t \leftarrow t'}$, $\mathbf{X}^{t';t \leftarrow t'}$) into the same global coordinate frame to produce world-coordinate pointmap $\mathbf{X}^t \in \mathbb{R}^{H \times W \times 3}$. To do this, as shown in Fig. 3, we use DUSt3R's alignment loss and add two video specific loss terms: camera trajectory smoothness and flow projection.

We start by re-parameterizing the global pointmaps $\mathbf{X}^t$ with camera parameters $\mathbf{P}^t = [\mathbf{R}^t | \mathbf{T}^t], \mathbf{K}^t$ and per-frame depthmap $\mathbf{D}^t$, as $\mathbf{X}_{i,j}^t := \mathbf{P}^{t-1} h(\mathbf{K}^{t-1}[i\mathbf{D}_{i,j}^t; j\mathbf{D}_{i,j}^t; \mathbf{D}_{i,j}^t])$, with $(i, j)$ for pixel coordinate and $h()$ for homogeneous mapping. It allows us to define losses directly on the camera parameters. To simplify the notation for function parameters, we use $\mathbf{X}^t$ as a shortcut for $\mathbf{P}^t, \mathbf{K}^t, \mathbf{D}^t$.

First, we use the alignment term in DUSt3R which aims to find a single rigid transformation $\mathbf{P}^{t;e}$ that aligns each pairwise estimation with the world coordinate pointmaps, since both $\mathbf{X}^{t;t \leftarrow t'}$ and $\mathbf{X}^{t';t \leftarrow t'}$ are in the same camera coordinate frame:

$$\mathcal{L}_{\mathrm{align}}(\mathbf{X}, \sigma, \mathbf{P}_W) = \sum_{W^i \in W} \sum_{e \in W^i} \sum_{t \in e} ||\mathbf{C}^{t;e} \cdot (\mathbf{X}^t - \sigma^e \mathbf{P}^{t;e} \mathbf{X}^{t;e})||_1, \tag{4}$$

where $\sigma^e$ is a pairwise scale factor. To simplify the notation, we use the directed edge $e = (t, t')$ interchangeably with $t \leftarrow t'$.

We use a camera trajectory smoothness loss to encourage smooth camera motion by penalizing large changes in camera rotation and translation in nearby timesteps:

$$\mathcal{L}_{\text{smooth}}(\mathbf{X}) = \sum_{t=0}^{N} \left( \left\| \mathbf{R}^{t\top} \mathbf{R}^{t+1} - \boldsymbol{I} \right\|_{\text{f}} + \left\| \mathbf{T}^{t+1} - \mathbf{T}^{t} \right\|_{2} \right), \tag{5}$$

where the Frobenius norm $\| \cdot \|_{\text{f}}$ is used for the rotation difference, the L2 norm $\| \cdot \|_{2}$ is used for the translation difference, and $\boldsymbol{I}$ is the identity matrix.

We also use a flow projection loss to encourage the global pointmaps and camera poses to be consistent with the estimated flow for the confident, static regions of the actual frames. More precisely, given two frames $t, t'$, using their global pointmaps, camera extrinsics and intrinsics, we compute the flow fields from taking the global pointmap $\mathbf{X}^{t}$, assuming the scene is static, and then moving the camera from $t$ to $t'$. We denote this value $\mathbf{F}_{\text{cam}}^{\text{global};t\to t'}$, similar to the term defined in the confident static region computation above. Then we can encourage this to be close to the estimated flow, $\mathbf{F}_{\text{est}}^{t\to t'}$, in the regions which are confidently static $\mathbf{S}^{\text{global};t\to t'}$ according to the global parameters:

$$\mathcal{L}_{\text{flow}}(\mathbf{X}) = \sum_{W^i \in W} \sum_{t\to t' \in W^i} || \mathbf{S}^{\text{global};t\to t'} \cdot (\mathbf{F}_{\text{cam}}^{\text{global};t\to t'} - \mathbf{F}_{\text{est}}^{t\to t'}) ||_1, \tag{6}$$

where $\cdot$ indicates element-wise multiplication. Note that the confident static mask is initialized using the pairwise prediction values (pointmaps and relative poses) as described in Sec. 3.3. During the optimization, we use the global pointmaps and camera parameters to compute $\mathbf{F}_{\text{cam}}^{\text{global}}$ and update the confident static mask. Please refer to Appendix D for more details on $\mathcal{L}_{\text{smooth}}$ and $\mathcal{L}_{\text{flow}}$.

The complete optimization for our dynamic global point cloud and camera poses is:

$$\hat{\mathbf{X}} = \underset{\mathbf{X}, \mathbf{P}_W, \sigma}{\arg\min} \mathcal{L}_{\text{align}}(\mathbf{X}, \sigma, \mathbf{P}_W) + w_{\text{smooth}} \mathcal{L}_{\text{smooth}}(\mathbf{X}) + w_{\text{flow}} \mathcal{L}_{\text{flow}}(\mathbf{X}), \tag{7}$$

where $w_{\text{smooth}}, w_{\text{flow}}$ are hyperparameters. Note, based on the reparameterization above, $\hat{\mathbf{X}}$ includes all the information for $\hat{\mathbf{D}}, \hat{\mathbf{P}}, \hat{\mathbf{K}}$.

**Video depth.** We can now easily obtain temporally-consistent video depth, traditionally addressed as a standalone problem. Since our global pointmaps are parameterized by camera pose and per-frame depthmaps $\hat{\mathbf{D}}$, just returning $\hat{\mathbf{D}}$ gives the video depth.

## 4 EXPERIMENTS

MonST3R runs on a monocular video of a dynamic scene and jointly optimizes video depth and camera pose. We compare the performance with methods specially designed for each individual subtask (*i.e.*, depth estimation and camera pose estimation), as well as monocular depth methods.

### 4.1 EXPERIMENTAL DETAILS

**Training and Inference.** We fine-tune the DUSt3R's ViT-Base decoder and DPT heads for 25 epochs, using 20,000 sampled image pairs per epoch. We use the AdamW optimizer with a learning rate of $5 \times 10^{-5}$ and a mini-batch size of 4 per GPU. Training took one day on $2\times$ RTX 6000 48GB GPUs. Inference for a 60-frame video with $w = 9$ and stride 2 (approx. 600 pairs) takes around 30s.

**Global Optimization.** For global optimization Eq. (7), we set the hyperparameter of each weights to be $w_{\text{smooth}} = 0.01$ and $w_{\text{flow}} = 0.01$. We only enable the flow loss when the average value is below 20, when the poses are roughly aligned. The motion mask is updated during optimization if the per-pixel flow loss is higher than 50. We use the Adam optimizer for 300 iterations with a learning rate of 0.01, which takes around 1 minute for a 60-frame video on a single RTX 6000 GPU.

### 4.2 SINGLE-FRAME AND VIDEO DEPTH ESTIMATION

**Baselines.** We compare our method with video depth methods, NVDS (Wang et al., 2023), ChronoDepth (Shao et al., 2024), and concurrent work, DepthCrafter (Hu et al., 2024), as well as single-frame depth methods, Depth-Anything-V2 (Yang et al., 2024b) and Marigold (Ke et al., 2024).

Table 2: **Video depth evaluation** on Sintel, Bonn, and KITTI datasets. We evaluate for both scale-and-shift-invariant and scale-invariant depth. The best and second best results in each category are **bold** and underlined, respectively.

| Alignment | Category | Method | Sintel | | Bonn | | KITTI | |
|---|---|---|---|---|---|---|---|---|
| | | | Abs Rel $\downarrow$ | $\delta<1.25\uparrow$ | Abs Rel $\downarrow$ | $\delta<1.25\uparrow$ | Abs Rel $\downarrow$ | $\delta<1.25\uparrow$ |
| Per-sequence scale & shift | Single-frame depth | Marigold | 0.532 | 51.5 | 0.091 | 93.1 | 0.149 | 79.6 |
| | | Depth-Anything-V2 | 0.367 | 55.4 | 0.106 | 92.1 | 0.140 | 80.4 |
| | Video depth | NVDS | 0.408 | 48.3 | 0.167 | 76.6 | 0.253 | 58.8 |
| | | ChronoDepth | 0.687 | 48.6 | 0.100 | 91.1 | 0.167 | 75.9 |
| | | DepthCrafter (Sep. 2024) | **0.292** | **69.7** | 0.075 | **97.1** | 0.110 | 88.1 |
| | Joint video depth & pose | Robust-CVD | 0.703 | 47.8 | - | - | - | - |
| | | CasualSAM | 0.387 | 54.7 | 0.169 | 73.7 | 0.246 | 62.2 |
| | | **MonST3R** | 0.335 | 58.5 | **0.063** | 96.4 | **0.104** | **89.5** |
| Per-sequence scale | Video depth | DepthCrafter (Sep. 2024) | 0.692 | 53.5 | 0.217 | 57.6 | 0.141 | 81.8 |
| | Joint depth & pose | **MonST3R** | **0.345** | **56.2** | **0.065** | **96.3** | **0.106** | **89.3** |

Table 3: **Single-frame depth evaluation.** We report the performance on Sintel, Bonn, KITTI, and NYU-v2 (static) datasets. MonST3R achieves overall comparable results to DUSt3R.

| Method | Sintel | | Bonn | | KITTI | | NYU-v2 (static) | |
|---|---|---|---|---|---|---|---|---|
| | Abs Rel $\downarrow$ | $\delta<1.25\uparrow$ | Abs Rel $\downarrow$ | $\delta<1.25\uparrow$ | Abs Rel $\downarrow$ | $\delta<1.25\uparrow$ | Abs Rel $\downarrow$ | $\delta<1.25\uparrow$ |
| DUSt3R | 0.424 | 58.7 | 0.141 | 82.5 | 0.112 | 86.3 | 0.080 | 90.7 |
| **MonST3R** | 0.345 | 56.5 | 0.076 | 93.9 | 0.101 | 89.3 | 0.091 | 88.8 |

We also compare with methods for joint video depth and pose estimation, CasualSAM (Zhang et al., 2022) and Robust-CVD (Kopf et al., 2021), which address the same problem as us. This comparison is particularly important since joint estimation is substantially more challenging than only estimating depth. Of note, CasualSAM relies on heavy optimization, whereas ours runs in a feed-forward manner with only lightweight optimization.

**Benchmarks and metrics.** Similar to DepthCrafter, we evaluate video depth on KITTI (Geiger et al., 2013), Sintel (Butler et al., 2012), and Bonn (Palazzolo et al., 2019) benchmark datasets, covering dynamic and static, indoor and outdoor, and realistic and synthetic data. For monocular/single-frame depth estimation, we also evaluate on NYU-v2 (Silberman et al., 2012).

Our evaluation metrics include absolute relative error (Abs Rel) and percentage of inlier points $\delta < 1.25$, following the convention (Hu et al., 2024; Yang et al., 2024b). All methods output scale-and/or shift- invariant depth estimates. For video depth evaluation, we align a single scale and/or shift factor per each sequence, whereas the single-frame evaluation adopts per-frame median scaling, following Wang et al. (2024c). As demonstrated by Yin et al. (2021), shift is particularly important in the 3D geometry of a scene and is important to predict.

**Results.** As shown in Tab. 2, MonST3R achieves competitive and even better results, even outperforming specialized video depth estimation techniques like DepthCrafter (a concurrent work). Furthermore, MonST3R significantly outperforms DepthCrafter (Hu et al., 2024) with scale-only normalization. As in Tab. 3, even after our fine-tuning for videos of dynamic scenes, the performance on single-frame depth estimation remains competitive with the original DUSt3R model.

## 4.3 CAMERA POSE ESTIMATION

**Baselines.** We compare with not only direct competitors (*i.e.*, CasualSAM and Robust-CVD), but also a range of learning-based visual odometry methods for dynamic scenes, such as DROID-SLAM (Teed & Deng, 2021), Particle-SfM (Zhao et al., 2022), DPVO (Teed et al., 2024), and LEAP-VO (Chen et al., 2024). Notably, several methods (*e.g.*, DROID-SLAM, DPVO, and LEAP-VO) require ground truth camera intrinsic as input and ParticleSfM is an optimization-based method that runs $5\times$ slower than ours. We also compare with the "DUSt3R with mask" baseline in Sec. 3.1 to see if DUSt3R performs well on dynamic scenes when a ground truth motion mask is provided.

**Benchmarks and metrics.** We evaluate the methods on Sintel (Butler et al., 2012) and TUM-dynamics (Sturm et al., 2012) (following CasualSAM) and ScanNet (Dai et al., 2017) (following

Table 4: **Evaluation on camera pose estimation** on the Sintel, TUM-dynamic, and ScanNet. The best and second best results are **bold** and underlined, respectively. MonST3R achieves competitive and even better results than pose-specific methods, even without ground truth camera intrinsics.

| Category | Method | Sintel | | | TUM-dynamics | | | ScanNet (static) | | |
|---|---|---|---|---|---|---|---|---|---|---|
| | | ATE ↓ | RPE trans ↓ | RPE rot ↓ | ATE ↓ | RPE trans ↓ | RPE rot ↓ | ATE ↓ | RPE trans ↓ | RPE rot ↓ |
| Pose only | DROID-SLAM* | 0.175 | 0.084 | 1.912 | - | - | - | - | - | - |
| | DPVO* | 0.115 | 0.072 | 1.975 | - | - | - | - | - | - |
| | ParticleSfM | 0.129 | **0.031** | **0.535** | - | - | - | 0.136 | 0.023 | 0.836 |
| | LEAP-VO* | **0.089** | 0.066 | 1.250 | 0.046 | 0.027 | **0.385** | 0.070 | 0.018 | **0.535** |
| Joint depth & pose | Robust-CVD | 0.360 | 0.154 | 3.443 | 0.189 | 0.071 | 3.681 | 0.227 | 0.064 | 7.374 |
| | CasualSAM | 0.141 | 0.035 | 0.615 | **0.045** | 0.020 | 0.841 | 0.158 | 0.034 | 1.618 |
| | DUSt3R w/ mask[†] | 0.417 | 0.250 | 5.796 | 0.127 | 0.062 | 3.099 | 0.081 | 0.028 | 0.784 |
| | **MonST3R** | 0.108 | 0.042 | 0.732 | 0.074 | **0.019** | 0.905 | **0.068** | **0.017** | 0.545 |

[*] requires ground truth camera intrinsics as input, [†] unable to estimate the depth of foreground object.

ParticleSfM) to test generalization to static scenes as well. On Sintel, we follow the same evaluation protocol as in Chen et al. (2024); Zhao et al. (2022), which excludes static scenes or scenes with perfectly-straight camera motion, resulting in total 14 sequences. For TUM-dynamics and ScanNet, we sample the first 90 frames with the temporal stride of 3 to save compute. We report the same metric as Chen et al. (2024); Zhao et al. (2022): Absolute Translation Error (ATE), Relative Translation Error (RPE trans), and Relative Rotation Error (RPE rot), after applying a Sim(3) Umeyama alignment on prediction to the ground truth.

**Results.** In Tab. 4, MonST3R achieves the best accuracy in Sintel and ScanNet among methods to joint depth and pose estimation and performs competitively to pose-only methods, even without using ground truth camera intrinsics. Our method also generalizes well to static scenes (*i.e.*, Scan-Net) and shows improvements over even DUSt3R, which proves the effectiveness of our designs (*e.g.*, Eq. (7)) for video input.

### 4.4 JOINT DENSE RECONSTRUCTION AND POSE ESTIMATION

Fig. 4 qualitatively compares our method with CasualSAM and DUSt3R on video sequences for joint dense reconstruction and pose estimation on DAVIS (Perazzi et al., 2016). For each video sequence, we visualize overlayed point clouds aligned with estimated camera pose, showing as two rows with different view points for better visualization. As discussed in Fig. 2, DUSt3R struggles with estimating correct geometry of moving foreground objects, resulting in failure of joint camera pose estimation and dense reconstruction. CasualSAM reliably estimates camera trajectories while sometimes failing to produce correct geometry estimates for foreground objects. MonST3R outputs both reliable camera trajectories and reconstruction of entire scenes along the video sequences.

### 4.5 ABLATION STUDY

Table 5 presents an ablation study analyzing the impact of design choices in our method, including the selection of training datasets, fine-tuning strategies, and the novel loss functions used for dynamic point cloud optimization. Our analysis reveals that: (1) all datasets contribute to improved camera pose estimation performance; (2) fine-tuning only the decoder and head outperforms alternative strategies; and (3) the proposed loss functions enhance pose estimation with minimal impact on video depth accuracy.

**Discussions.** While MonST3R represents a promising step towards directly estimating dynamic geometry from videos as well as camera pose and video depth, limitations remain. While our method can, unlike prior methods, theoretically handle dynamic camera intrinsics, we find that, in practice, this requires careful hyperparameter tuning or manual constraints. To trade off compute and performance, our global alignment applies a relatively small size of the sliding window, making it vulnerable to long-term occlusion. Additionally, Like many deep learning methods, MonST3R struggles with out-of-distribution inputs, such as open fields. Expanding the training set is a key direction to make MonST3R more robust to in-the-wild videos.

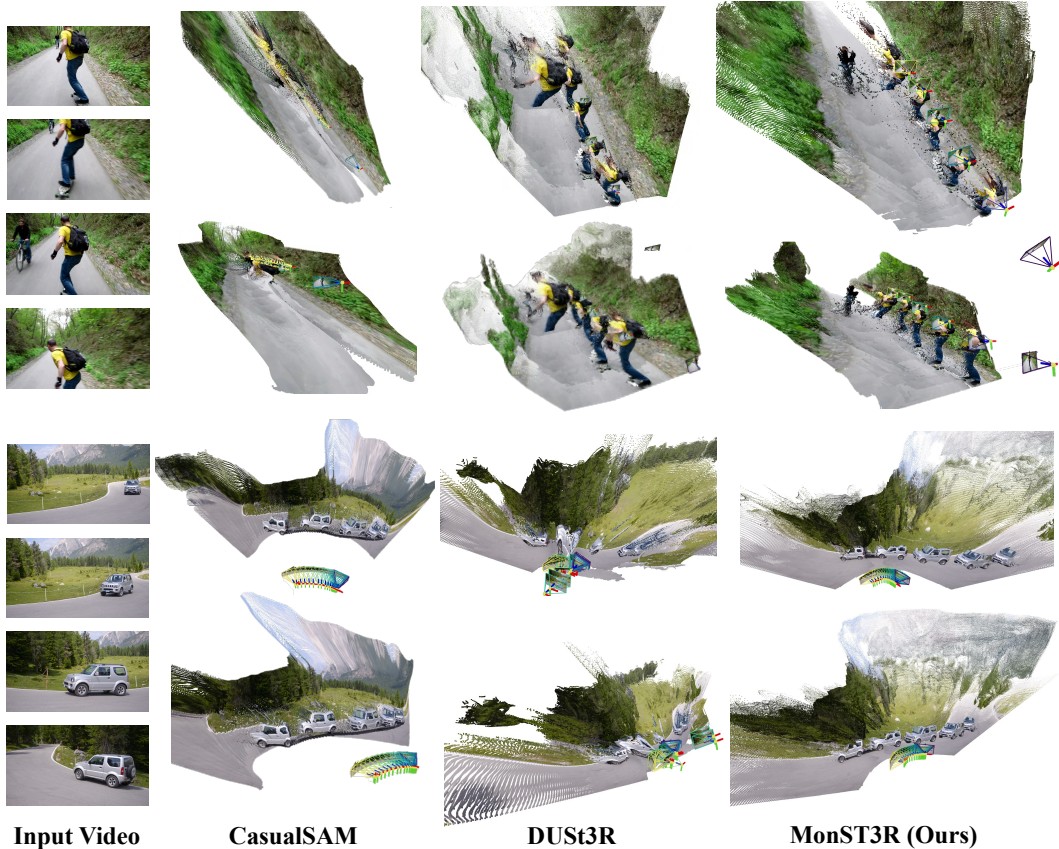

| Input Video | CasualSAM | DUSt3R | MonST3R (Ours) |

Figure 4: **Qualitative comparison.** Compared to CasualSAM and DUSt3R, our method outputs both reliable camera trajectories and geometry of dynamic scenes. Refer to Fig. A5 for more results.

Table 5: **Ablation study on Sintel dataset.** For each category, the default setting is underlined, and the best performance is **bold**.

| | Variants | Camera pose estimation | | | Video depth estimation | |
|---|---|---|---|---|---|---|
| | | ATE ↓ | RPE trans ↓ | RPE rot ↓ | Abs Rel ↓ | δ <1.25 ↑ |
| **Training dataset** | No finetune (DUSt3R weights) | 0.354 | 0.167 | 0.996 | 0.482 | 56.5 |
| | w/ PO | 0.220 | 0.129 | 0.901 | 0.378 | 53.7 |
| | w/ PO+TA | 0.158 | 0.054 | 0.886 | 0.362 | 56.7 |
| | w/ PO+TA+Spring | 0.121 | 0.046 | 0.777 | **0.329** | 58.1 |
| | w/ TA+Spring+Waymo | 0.167 | 0.107 | 1.136 | 0.462 | 54.0 |
| | w/ all 4 datasets | **0.108** | **0.042** | **0.732** | 0.335 | **58.5** |
| **Training strategy** | Full model finetune | 0.181 | 0.110 | 0.738 | 0.352 | 55.4 |
| | Finetune decoder & head | **0.108** | **0.042** | **0.732** | **0.335** | **58.5** |
| | Finetune head | 0.185 | 0.128 | 0.860 | 0.394 | 55.7 |
| **Inference** | w/o flow loss | 0.140 | 0.051 | 0.903 | 0.339 | 57.7 |
| | w/o static region mask | 0.132 | 0.049 | 0.899 | 0.334 | **58.7** |
| | w/o smoothness loss | 0.127 | 0.060 | 1.456 | **0.333** | 58.4 |
| | Full | **0.108** | **0.042** | **0.732** | 0.335 | 58.5 |

## 5 CONCLUSIONS

We present MonST3R, a simple approach to directly estimate geometry for dynamic scenes and extract downstream information like camera pose and video depth. MonST3R leverages per-timestep pointmaps as a powerful representation for dynamic scenes. Despite being finetuned on a relatively small training dataset, MonST3R achieves impressive results on downstream tasks, surpassing even state-of-the-art specialized techniques.

ACKNOWLEDGMENTS

We would like to express gratitude to Youming Deng, Hang Gao, and Haven Feng for their helpful discussions and extend special thanks to Brent Yi, Chung Min Kim, and Justin Kerr for their valuable help with online visualizations. This work is supported by the Intelligence Advanced Research Projects Activity (IARPA) via Department of Interior/ Interior Business Center (DOI/IBC) contract number 140D0423C0074. The U.S. Government is authorized to reproduce and distribute reprints for Governmental purposes notwithstanding any copyright annotation thereon. Disclaimer: The views and conclusions contained herein are those of the authors and should not be interpreted as necessarily representing the official policies or endorsements, either expressed or implied, of IARPA, DOI/IBC, or the U.S. Government.

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

## A MORE QUALITATIVE RESULTS

### A.1 DEPTH

For more thorough comparisons, we include additional qualitative examples of video depth, comparing our method against DepthCrafter, a concurrent method specifically trained for video depth. We include comparisons on the Bonn dataset in Fig. A1 and the KITTI dataset in Fig. A2. In these comparisons, we show that after alignment, our estimates are much closer to the ground truth than those of DepthCrafter.

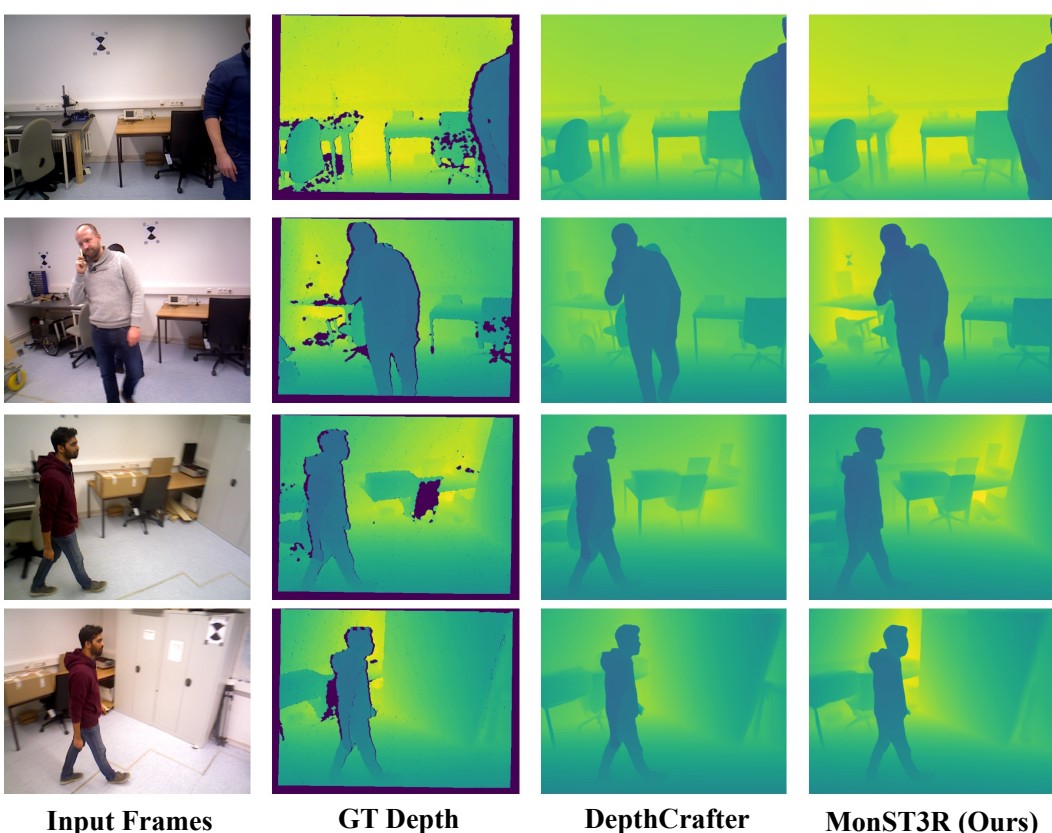

| **Input Frames** | **GT Depth** | **DepthCrafter** | **MonST3R (Ours)** |

Figure A1: **Video depth estimation comparison on Bonn dataset.** Evaluation protocol is persequence scale & shift. We visualize the prediction result *after* alignment. Note, in the first row, our depth estimation is more aligned with the GT depth (*e.g.*, the wall) compared to DepthCrafter's.

### A.2 CAMERA POSE

We present additional qualitative results for camera pose estimation. We compare our model with the state-of-the-art visual odometry method LEAP-VO and the joint video depth and pose optimization method CasualSAM. Results are provided for the Sintel dataset in Fig. A3 and Scannet dataset in Fig. A4. In these comparisons, our method significantly outperforms the baselines for very challenging cases such as "temple_3" and "cave_2" in Sintel and performs comparable to or better than baselines in the rest of the results like those in the Scannet dataset.

### A.3 JOINT DEPTH & CAMERA POSE RESULTS

We present additional results for joint point cloud and camera pose estimation, comparing against CasualSAM and DUSt3R. Fig. A5 shows three additional scenes for Davis: mbike-trick, train, and dog. For mbike-trick, CasualSAM makes large errors in both geometry and camera pose; DUSt3R produces reasonable geometry except for the last subset of the video which also results in poor

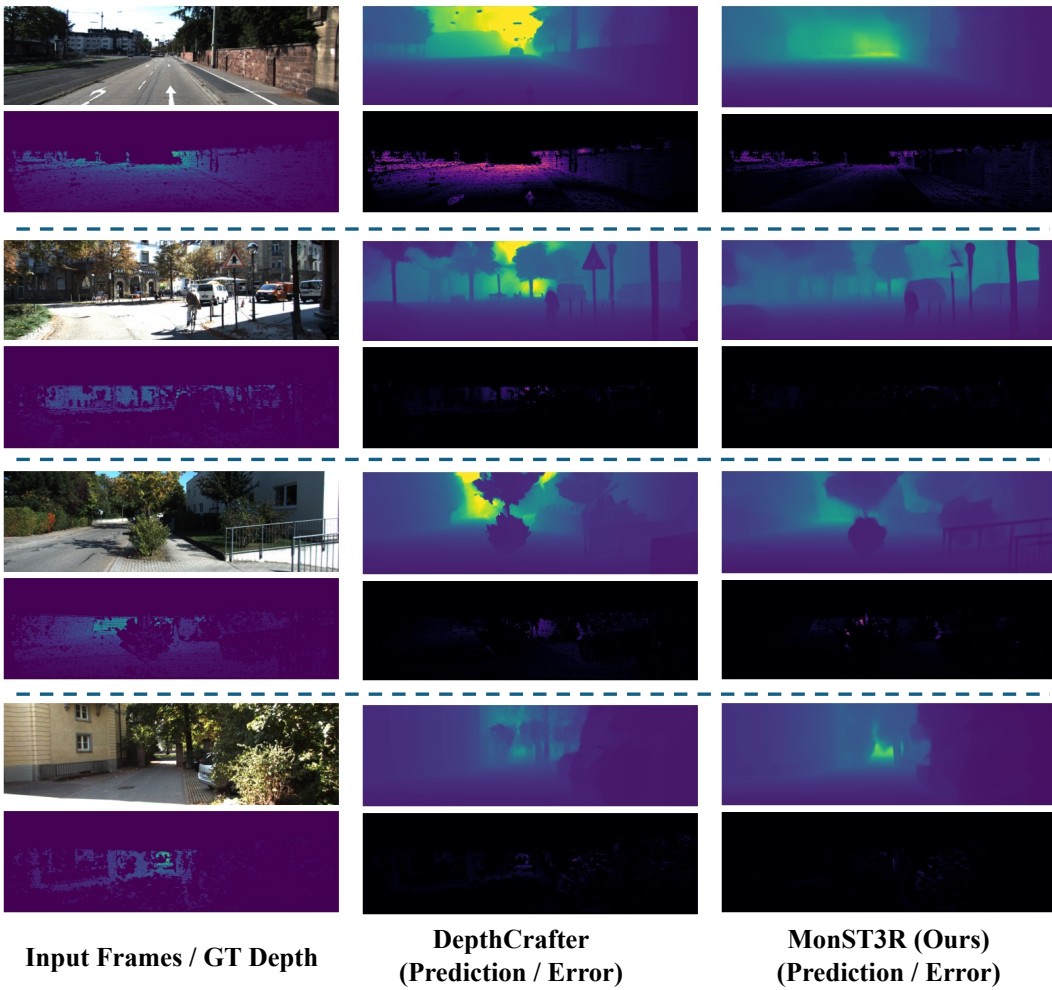

**Input Frames / GT Depth**      **DepthCrafter (Prediction / Error)**      **MonST3R (Ours) (Prediction / Error)**

Figure A2: **Video depth estimation comparison on KITTI dataset.** Evaluation protocol is per-sequence scale & shift. For each case, the upper row is for input frame and depth prediction; the lower row is for ground truth depth annotation and error map. Prediction result is *after* alignment.

pose estimation (highlighted in red); and ours correctly estimates both geometry and camera pose. For train, CasualSAM accurately recovers the camera pose for the video but produces suboptimal geometry, misaligning the train and the track at the top right. DUSt3R both misaligns the track at the top left and gives poor camera pose estimates. Our method correctly estimates both geometry and camera pose. For dog, CasualSAM produces imprecise, smeared geometry with slight inaccuracies in the camera pose. DUSt3R results in mistakes in both the geometry and camera pose due to misalignments of the frames, while our method correctly estimates both geometry and camera pose.

## A.4 PAIRWISE POINTMAPS

In Fig. A6, we also include visualizations of two input frames and estimated pairwise pointmaps, the direct output of the trained models, for both DUSt3R and MonST3R. Note, these results do not include any optimization or post-processing. Row 1 demonstrates that even after fine-tuning, our method retains the ability to handle changing camera intrinsics. Rows 2 and 3 demonstrate that our method can handle "impossible" alignments that two frames have almost no overlap, even in the presence of motion, unlike DUSt3R that misaligns based on the foreground object. Rows 4 and 5 show that in addition to enabling the model to handle motion, our fine-tuning also has improved the model's ability to represent large-scale scenes, where DUSt3R predicts to be flat.

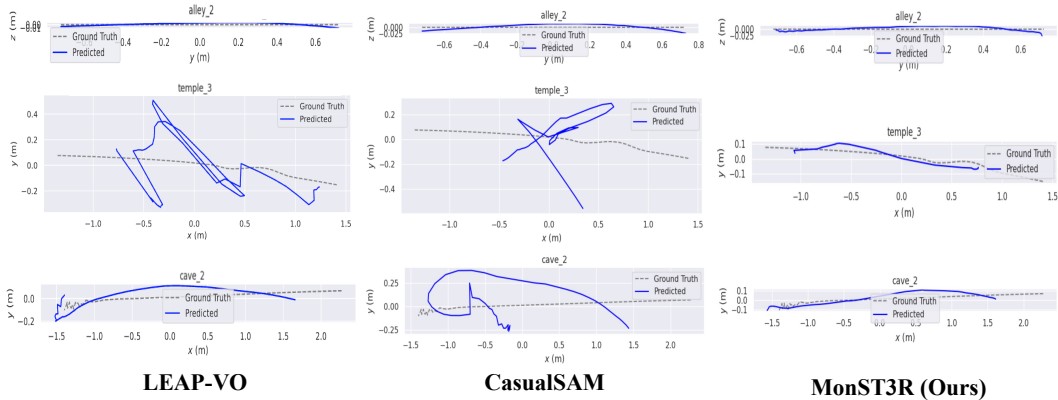

Figure A3: **Camera pose estimation comparison on the Sintel dataset.** The trajectories are plotted along the two axes with the highest variance to capture the most significant motion. The predicted trajectory (solid blue line) is aligned to match the ground truth trajectory (dashed gray line). Our MonST3R is more robust at challenging scenes, "temple_3" and "cave_2" (the last two rows).

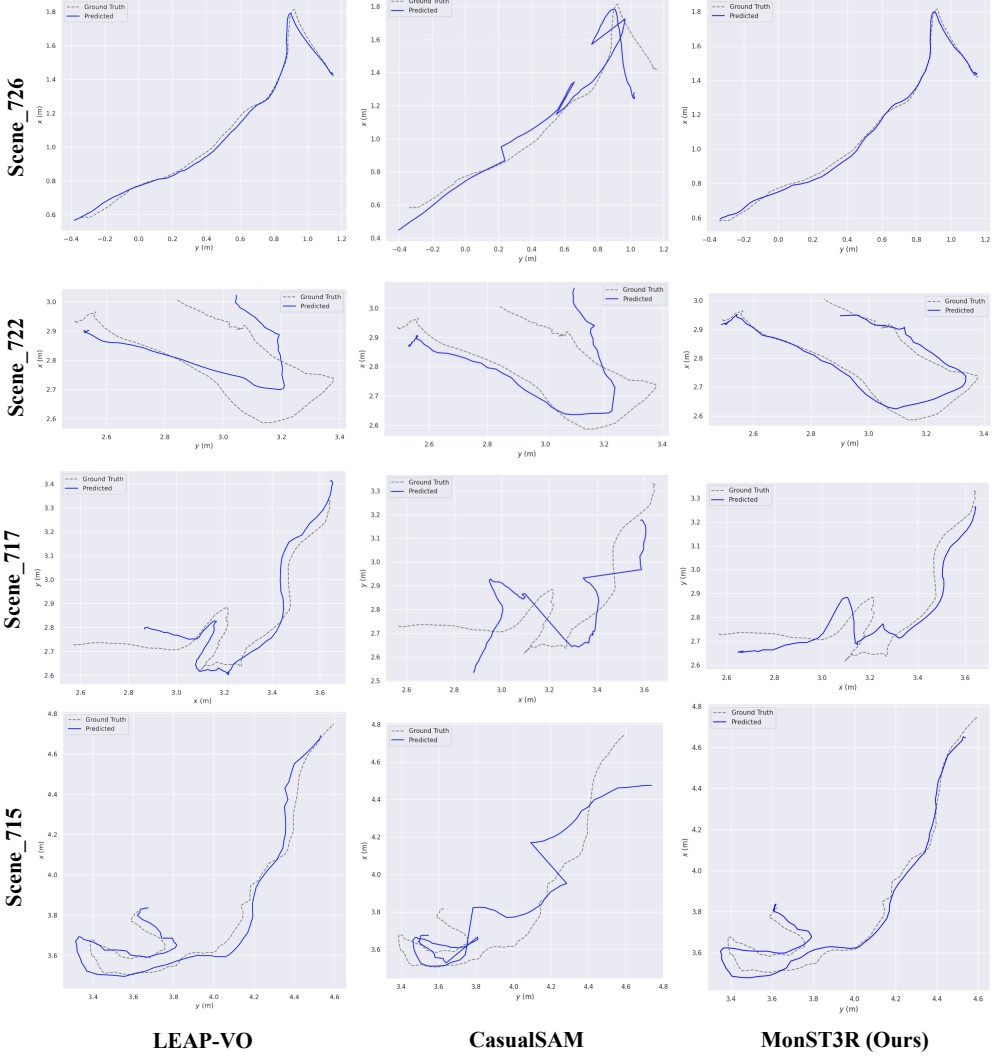

Figure A4: **Camera pose estimation comparison on the Scannet dataset.** The trajectories are plotted along the two axes with the highest variance to capture the most significant motion.

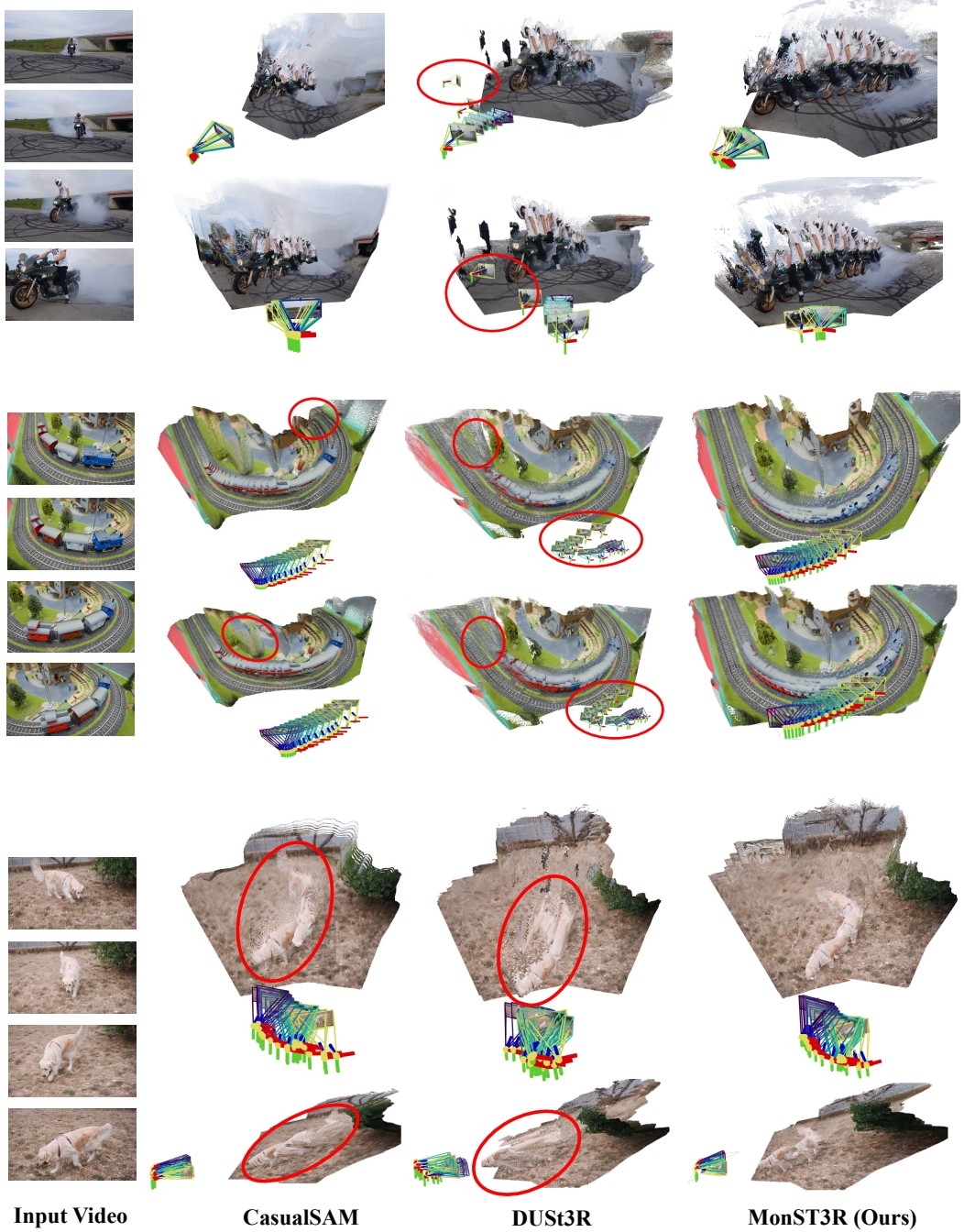

Figure A5: **Qualitative comparison on Davis.** Compared to CasualSAM and DUSt3R, our method outputs both reliable camera trajectories and geometry of dynamic scenes.

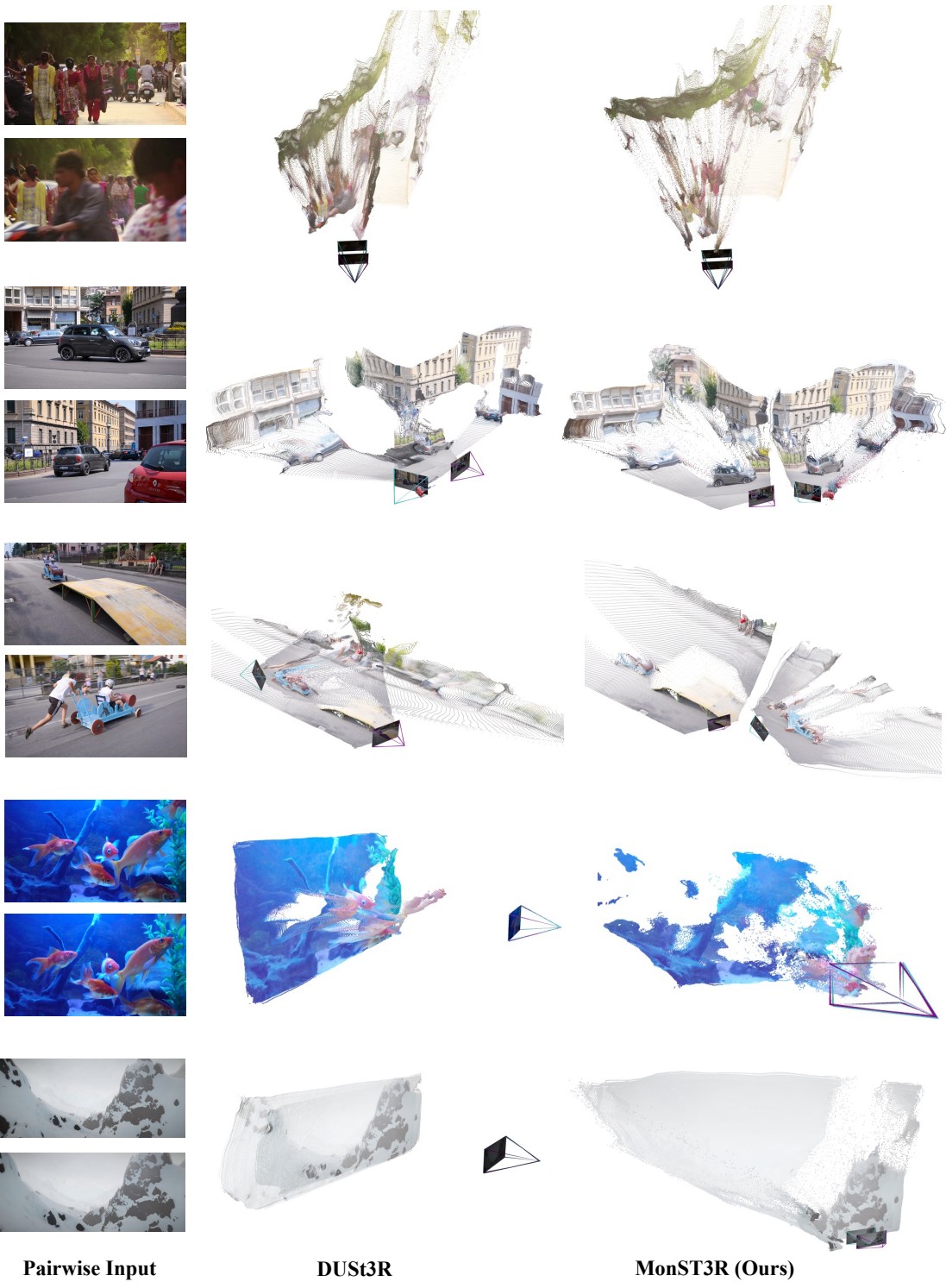

Figure A6: **Qualitative comparison of feed-forward pairwise pointmaps prediction.**

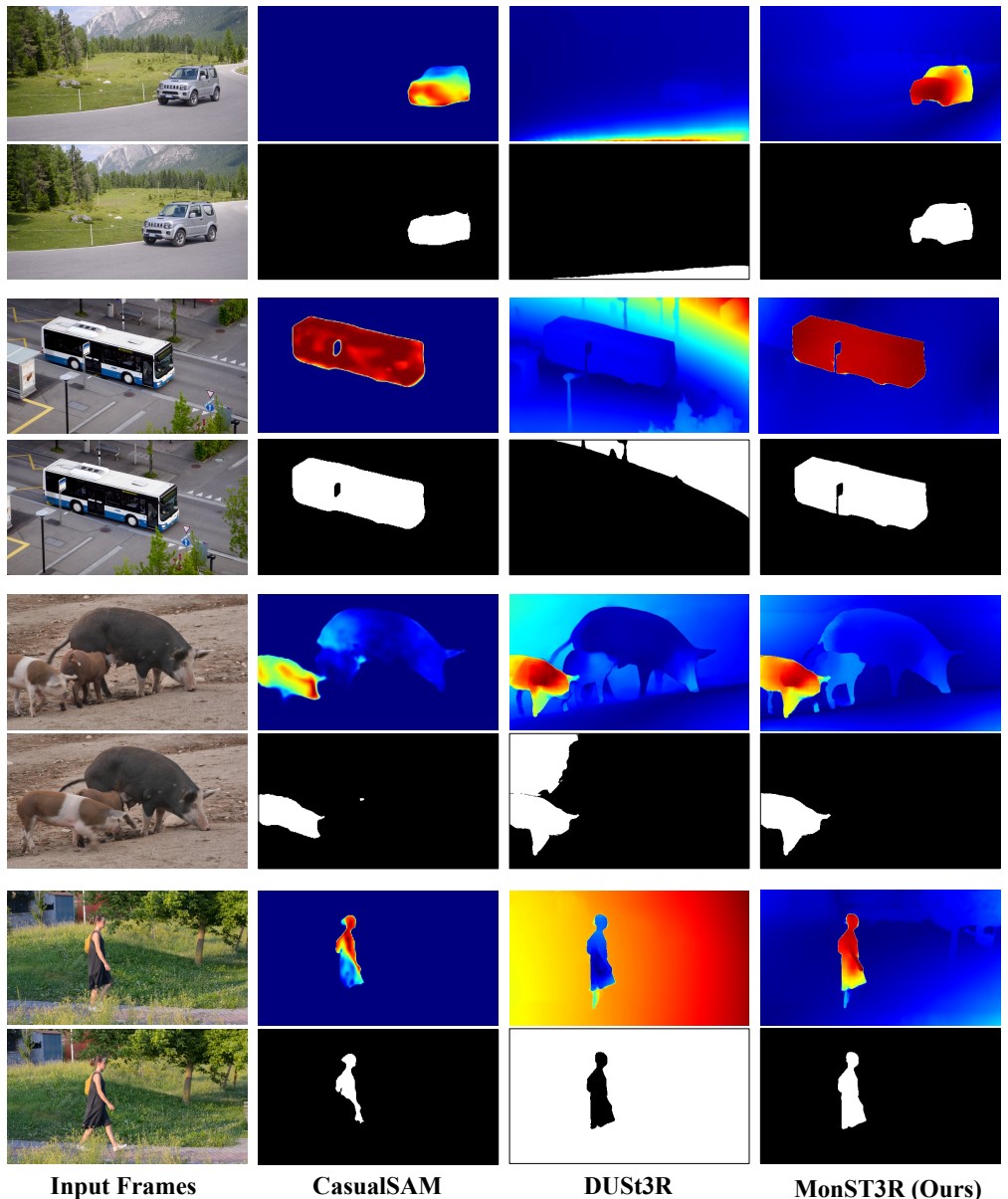

| Input Frames | CasualSAM | DUSt3R | MonST3R (Ours) |

Figure A7: **Qualitative comparison of static/dynamic mask.** We visualize both the continuous error map (upper row) and binary static/dynamic mask (lower row). The threshold $\alpha$ is fixed.

## A.5 STATIC/DYNAMIC MASK

We present the result of static/dynamic mask estimation based on the optical flow, as discussed in Sec. 3.3. As shown in Fig. A7, our MonST3R achieves overall plausible results while DUSt3R fails to provide accurate motion mask due to error in camera pose and depth estimation.

## B MORE QUANTITATIVE RESULTS

### B.1 ABLATION ON TRANING/INFERENCE WINDOW SIZE

In this section, we present additional quantitative results on the impact of different training and inference window sizes, as shown in Tab. A1. The results demonstrate that performance generally improves as the inference window size increases, up to the size of the training window.

Table A1: **Ablation study on different training/inference window sizes on the Sintel dataset.** Each cell displays two values: **ATE ↓ / Abs Rel ↓**, corresponding to camera pose and video depth estimation, respectively. The cells where the inference window size exceeds the training window size are highlighted in grey. The default setup is underlined, and the best results are in **bold**. GPU memory consumption for each inference setup is listed in the leftmost column.

| | | Training window size | | |
|---|---|---|---|---|
| Memory (GB) | Inference video graph | Size 5 | Size 7 | Size 9 |
| 17.3 | Window size 3 | 0.191 / 0.442 | 0.182 / 0.413 | 0.163 / 0.383 |
| 20.1 | Window size 4 | 0.178 / 0.431 | 0.166 / 0.406 | 0.148 / 0.362 |
| 17.2 | Window size 5 (stride 2) | 0.145 / 0.439 | 0.140 / 0.411 | 0.137 / 0.367 |
| 23.5 | Window size 5 | 0.139 / 0.406 | 0.132 / 0.399 | 0.133 / 0.355 |
| 19.9 | Window size 7 (stride 2) | 0.180 / 0.409 | 0.140 / 0.372 | 0.121 / 0.359 |
| 29.5 | Window size 7 | 0.174 / 0.389 | 0.136 / 0.351 | 0.113 / 0.346 |
| 23.2 | Window size 9 (stride 2) | 0.177 / 0.380 | 0.156 / 0.387 | **0.108 / 0.345** |

Moreover, our proposed stride-based sampling provides a better trade-off between window size and computational cost. For instance, for the training window size of 7 or 9, the inference configuration "Window size 7 (stride 2)" outperforms "Window size 4" while consuming a similar amount of memory (19.9 GB vs. 20.1 GB). Additionally, for a training window size of 9, "Window size 9 (stride 2)" achieves better performance than "Window size 7" while reducing memory consumption by 20%, highlighting the efficiency of our design.

## B.2 ABLATION ON LOSS WEIGHT SENSITIVITY

Table A2: **Ablation study on loss weight sensitivity.** The table shows the effect of varying the loss weights $w_{\text{smooth}}$ and $w_{\text{flow}}$ on camera pose and video depth estimation. The default setup is underlined, and the best results are in **bold**.

| | | Camera pose estimation | | | Video depth estimation | |
|---|---|---|---|---|---|---|
| $w_{\text{smooth}}$ | $w_{\text{flow}}$ | ATE ↓ | RPE$_{\text{trans}}$ ↓ | RPE$_{\text{rot}}$ ↓ | Abs Rel ↓ | $\delta < 1.25$ ↑ |
| 0.01 | 0.001 | 0.118 | 0.045 | 0.716 | 0.335 | 58.2 |
| 0.01 | 0.005 | 0.109 | **0.042** | **0.715** | 0.336 | 58.2 |
| 0.01 | 0.01 | **0.108** | **0.042** | 0.732 | 0.335 | 58.5 |
| 0.01 | 0.05 | 0.115 | 0.044 | 0.831 | **0.329** | **59.5** |
| 0.01 | 0.1 | 0.118 | 0.049 | 0.838 | 0.330 | 59.3 |
| 0.001 | 0.01 | 0.110 | 0.048 | 0.869 | **0.332** | 58.4 |
| 0.005 | 0.01 | 0.109 | 0.044 | 0.844 | 0.334 | 58.3 |
| 0.01 | 0.01 | **0.108** | **0.042** | **0.732** | 0.335 | **58.5** |
| 0.05 | 0.01 | 0.127 | 0.045 | 0.779 | 0.342 | 57.9 |
| 0.1 | 0.01 | 0.138 | 0.049 | 0.799 | 0.346 | 57.3 |

In the main paper, our global optimization objective (see Eq. (7)) is a combination of three different loss terms, controlled by two hyperparameters. Here, we evaluate the sensitivity of the results to variations in these two weights.

From the results in Tab. A2, it can be seen that the weight of the optical flow loss ($w_{\text{flow}}$) does not significantly impact the overall performance. Varying the flow loss weight sometimes leads to slightly better results in other metrics. The weight of the camera trajectory smoothness constraint ($w_{\text{smooth}}$) exhibits more noticeable effects. When it is set to a lower value, the difference in performance remains small, though the RPE performance drops noticeably. However, when the weight is set too high, performance degrades, likely due to the over-constraining of the camera trajectory.

## C    FULLY FEED-FORWARD RECONSTRUCTION

The MonST3R (or DUST3R) model predicts pairwise pointmaps in the coordinate frame of the first image, which can be seen as the anchor frame. To enable faster and fully feed-forward reconstruction from monocular video input, we construct image pairs that align all the $T$ frames to the same anchor frame (*e.g.*, the first frame or the middle frame), denoted as $\{t_{\text{anchor}} \leftarrow t \mid t \in 1, \ldots, T\}$. This alignment ensures that the predicted point cloud of each frame shares the same camera coordinate system as the anchor frame and can be treated as a global point cloud, *i.e.*, $\mathbf{X}^t = \mathbf{X}^{t;t_{\text{anchor}} \leftarrow t}$.

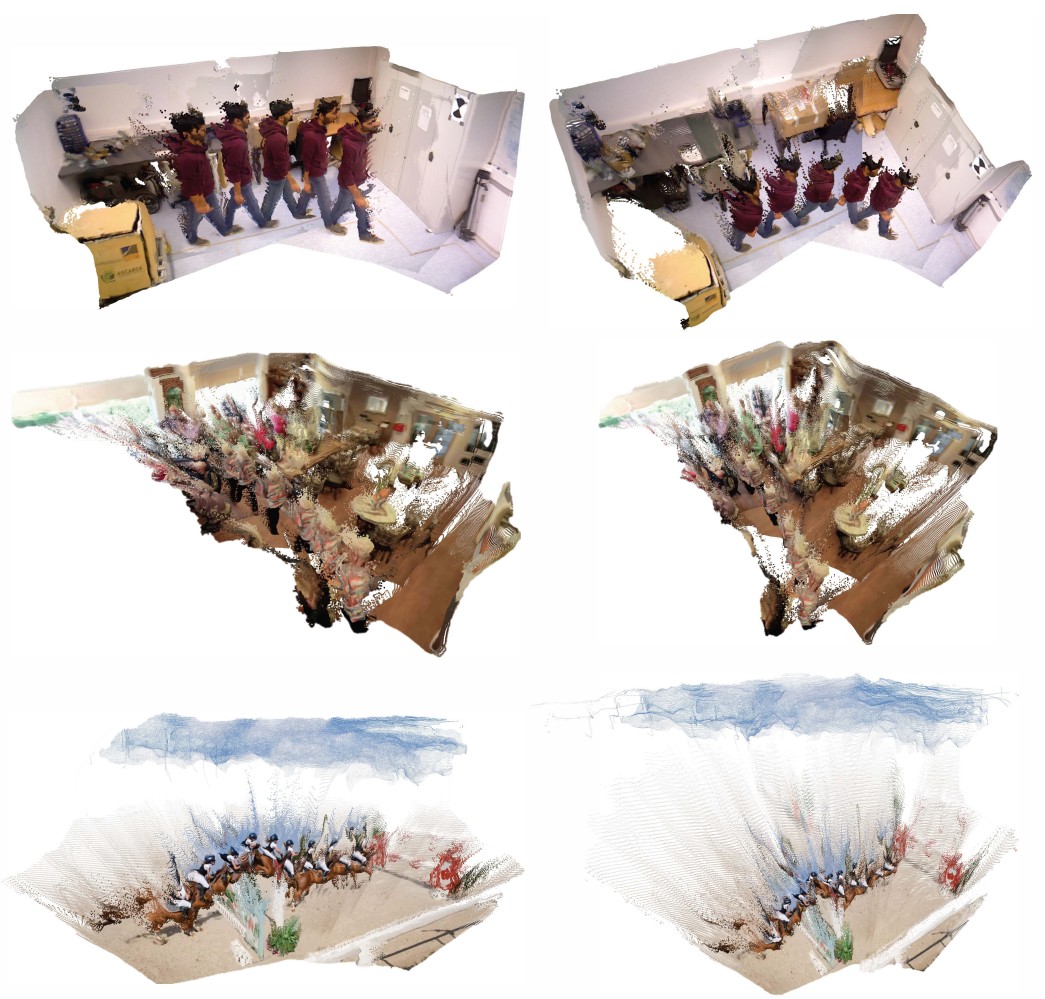

Figure A8: **Real-time reconstruction result.** By aligning all the frames to the middle frame, MonST3R enables real-time reconstruction from monocular video input. More video results at our webpage: https://monst3r-paper.github.io/page0.html.

This approach significantly enhances runtime performance, achieving approximately 40 FPS on a single RTX4090 GPU. Moreover, it has the potential to enable real-time reconstruction in a streaming fashion, by adaptively updating the anchor frame. We provide qualitative examples in Fig. A8.

Currently, this method has certain limitations. The reconstruction quality is sensitive to the choice of the anchor frame, and since each frame is aligned independently to the anchor, some artifacts (*e.g.*, shifting) may occur due to the lack of global information sharing. Nonetheless, we believe this approach is a promising direction for achieving streaming, real-time, and fully feed-forward reconstruction from monocular video input.

# D DETAILS ON GLOBAL OPTIMIZATION

## D.1 DETAILS ON $\mathcal{L}_{\text{SMOOTH}}$

The camera trajectory smoothness loss encourages smooth transitions between consecutive camera poses by penalizing large changes in rotation and translation. For frame $t$, given the rotation $\mathbf{R}^t$ and translation $\mathbf{T}^t$, the smoothness loss is defined as:

$$\mathcal{L}_{\text{smooth}}(\mathbf{R}, \mathbf{T}) = \sum_{t=0}^{N-1} \left( \left\| \mathbf{R}^{t\top} \mathbf{R}^{t+1} - \mathbf{I} \right\|_{\text{F}} + \left\| \mathbf{T}^{t+1} - \mathbf{T}^t \right\|_2 \right), \tag{8}$$

where $\mathbf{I}$ is the identity matrix, $\| \cdot \|_{\text{F}}$ denotes the Frobenius norm, and $\| \cdot \|_2$ denotes the Euclidean norm.

Since $\mathbf{R}$ and $\mathbf{T}$ parameterize the global point cloud $\mathbf{X}$ along with the depth map $\mathbf{D}$ and camera intrinsics $\mathbf{K}$, we simplify the notation by writing the smoothness loss as $\mathcal{L}_{\text{smooth}}(\mathbf{X}) := \mathcal{L}_{\text{smooth}}(\mathbf{R}, \mathbf{T})$ for brevity.

## D.2 DETAILS ON $\mathcal{L}_{\text{FLOW}}$

The flow projection loss ensures consistency between the camera-induced flow and the estimated optical flow for regions identified as static. It is defined as follows:

$$\mathcal{L}_{\text{flow}}(\mathbf{F}_{\text{cam}}, \mathbf{S}) = \sum_{W^i \in W} \sum_{t \rightarrow t' \in W^i} \left\| \mathbf{S}^{\text{global};t \rightarrow t'} \cdot \left( \mathbf{F}_{\text{cam}}^{\text{global};t \rightarrow t'} - \mathbf{F}_{\text{est}}^{t \rightarrow t'} \right) \right\|_1, \tag{9}$$

where $\mathbf{F}_{\text{cam}}^{\text{global};t \rightarrow t'}$ is the flow induced by camera motion from frame $t$ to frame $t'$, and $\mathbf{F}_{\text{est}}^{t \rightarrow t'}$ is the estimated optical flow obtained from an off-the-shelf method. The mask $\mathbf{S}^{\text{global};t \rightarrow t'}$ indicates regions that are confidently static.

The camera-induced flow $\mathbf{F}_{\text{cam}}^{\text{global};t \rightarrow t'}$ is computed using the global camera parameters, intrinsics, and depth map as follows:

$$\mathbf{F}_{\text{cam}}^{\text{global};t \rightarrow t'} = \pi \left( \mathbf{D}^t \mathbf{K}^{t'} \mathbf{R}^{t'} \mathbf{R}^{t\top} \mathbf{K}^{t-1} \hat{\mathbf{x}} + \mathbf{K}^{t'} (\mathbf{T}^{t'} - \mathbf{T}^t) \right) - \mathbf{x}, \tag{10}$$

where $\mathbf{x}$ is a pixel coordinate matrix, $\hat{\mathbf{x}}$ is $\mathbf{x}$ in homogeneous coordinates, and $\pi(\cdot)$ represents the projection operation from homogeneous to image coordinates.

To derive the confident static mask $\mathbf{S}^{\text{global};t \rightarrow t'}$, we first initialize a per-frame mask $\mathbf{S}^t$ with all the sampled pairs:

$$\mathbf{S}^t = \frac{1}{2|\mathcal{N}_t|} \left( \sum_{t' \in \mathcal{N}_t} \mathbf{S}^{t;t \rightarrow t'} + \sum_{t' \in \mathcal{N}_t} \mathbf{S}^{t;t' \rightarrow t} \right), \tag{11}$$

where $\mathcal{N}_t = \{t' \mid t \rightarrow t' \text{ is sampled}\}$. This initialization averages the pair-wise static masks from all sampled pairs involving frame $t$. For robustness, we also update the mask with global parameters, and derive the final mask as:

$$\mathbf{S}^{\text{global};t \rightarrow t'} = \mathbf{S}^t \vee \left[ \alpha > \left\| \mathbf{F}_{\text{cam}}^{\text{global};t \rightarrow t'} - \mathbf{F}_{\text{est}}^{t \rightarrow t'} \right\|_{\text{L1}} \right], \tag{12}$$

where $\vee$ denotes the logical "or" operator, and $\alpha$ is a predefined threshold.

Since both $\mathbf{F}_{\text{cam}}$ and $\mathbf{S}$ are derived from the global point cloud $\mathbf{X}$ (which includes $\mathbf{R}$, $\mathbf{T}$, $\mathbf{D}$, and $\mathbf{K}$), we express the flow loss as $\mathcal{L}_{\text{flow}}(\mathbf{X}) := \mathcal{L}_{\text{flow}}(\mathbf{F}_{\text{cam}}, \mathbf{S})$ for brevity.

