# OpenReview forum: "MonST3R: A Simple Approach for Estimating Geometry in the Presence of Motion"
_ICLR.cc/2025/Conference — ICLR 2025 Spotlight_

### Official Review · Reviewer_VTp6 · 2024-11-03

**Soundness:** 3
**Presentation:** 3
**Contribution:** 3
**Rating:** 6
**Confidence:** 5

**Summary:**

The paper introduces MonST3R, a novel geometry-first approach for estimating 3D structure from dynamic scenes in videos. Unlike existing methods that rely on multi-stage pipelines prone to errors, MonST3R adapts the DUSt3R pointmap representation, originally designed for static scenes, to handle moving objects by estimating pointmaps at each time step. This adaptation is achieved through fine-tuning on carefully selected synthetic and real-world datasets that provide depth and camera pose annotations, despite the challenge of limited data availability. The method demonstrates significant improvements in video depth and camera pose estimation, surpassing traditional optimization-based methods in robustness and efficiency. MonST3R is capable of generating reliable 4D reconstructions in a predominantly feed-forward manner, enhancing performance in various video-specific tasks.

**Strengths:**

1. The proposed problem of reconstructing static scenes and estimating per-timestep geometry of dynamic objects is compelling and relevant.
2. The paper presents a reasonable and well-structured solution to the defined problem.
3. The method achieves state-of-the-art performance, demonstrating its effectiveness.

**Weaknesses:**

1. The complete optimization in Eq. 6 minimizes the loss to obtain optimal values for X, P_W, and \sigma. However, it seems that the smoothness loss in Eq. 4 is minimized specifically for R and T. The input notation L_{smooth}(X) might be misleading—should it be L_{smooth}(R,T) instead?
2. The notation for L_{flow}(X) also seems ambiguous. Would L_{flow}(F) or L_{flow}(F,S) be more appropriate to reflect the actual inputs?
3. There is no explanation provided for the notation global. Clarifying what the global parameters are and how S^{global} is derived would be helpful. Specifically, is S^{global} in Eq. 5 an extension of Eq. 2 that uses F_{cam}^{global,t→t′} and F_{est}^{global,t→t′}
4. The proposed network is trained on diverse datasets, including PointOdyssey, TartanAir, Spring, and Waymo Perception, but the authors rely on a pretrained optical flow model. This could introduce a domain gap between the depth network and the optical flow network. Were there any issues related to domain gaps during the experiments?
5. It would be beneficial if the authors provided depth evaluation results for static scenes and dynamic objects separately. This would clarify whether the performance improvement is due to coherent depth estimation on static scenes or the enhanced depth quality of dynamic objects.
6. Do the authors use zero-shot models like Marigold and Depth-Anything? The use of the Waymo dataset (similar to KITTI) and the PointOdyssey dataset (with properties akin to Sintel) might explain why the proposed method outperforms these models. Can the authors specify which components of their approach are most critical for achieving this performance boost?
7. Overall, the technical parts is not clearly written.

**Questions:**

See the weakness part.

---

> ### Author Response · Authors · 2024-11-20
> **Official Comment by Authors (1/2)**
>
> **Q1-3: Clarification on the notation**
> 1. **The complete optimization in Eq. 6 minimizes the loss to obtain optimal values for X, P_W, and \sigma. However, it seems that the smoothness loss in Eq. 4 is minimized specifically for R and T. The input notation L_{smooth}(X) might be misleading—should it be L_{smooth}(R,T) instead?**
> 2. **The notation for L_{flow}(X) also seems ambiguous. Would L_{flow}(F) or L_{flow}(F,S) be more appropriate to reflect the actual inputs?**
> 3. **There is no explanation provided for the notation global. Clarifying what the global parameters are and how S^{global} is derived would be helpful. Specifically, is S^{global} in Eq. 5 an extension of Eq. 2 that uses F_{cam}^{global,t→t′} and F_{est}^{global,t→t′}**
>
> Thank you for your questions and suggestions. We have addressed these concerns as follows:
>
> 1./2.**Regarding $L_{\text{smooth}}(X)$ and $L_{\text{flow}}(X)$:**
>    As discussed in L313, $\mathbf{X}$ is reparameterized by $\mathbf{D}$ (depth), $\mathbf{K}$ (intrinsics), and $\mathbf{P}$ (camera pose, including $\mathbf{R}$ for rotation and $\mathbf{T}$ for translation). Since $\mathbf{R}, \mathbf{T}$ in $L_{\text{smooth}}$ and $\mathbf{F}, \mathbf{S}$ in $L_{\text{flow}}$ can be derived from $\mathbf{X}$, we use $\mathbf{X}$ in Eq. 4–6 for brevity. This is also clarified in **L316**, where we state: “To simplify the notation for function parameters, we use $\textbf{X}^t$ as a shortcut for $\mathbf{P}^t, \mathbf{K}^t, \mathbf{D}^t$.” Additional clarifications have been added in **Appendix Sec. B** of the revised version.
>
> 3.**Regarding the global notation:**
>    We have provided a detailed explanation of the global notation $F_{\text{cam}}^{\text{global}}$, in L333–L336 of the original paper. For the precise formulations of $F_{\text{cam}}^{\text{global}}$ and $S^{\text{global}}$, please refer to **Appendix Sec. B.2**, where we present these concepts in detail.
>
>
> ------
>
> **Q4: The proposed network is trained on diverse datasets, including PointOdyssey, TartanAir, Spring, and Waymo Perception, but the authors rely on a pretrained optical flow model. This could introduce a domain gap between the depth network and the optical flow network. Were there any issues related to domain gaps during the experiments?**
>
> Thank you for raising this point. While domain gaps can pose challenges, we observed that the pretrained optical flow model (SEA-RAFT) generalizes well across the datasets used in our experiments. This robustness is likely due to the nature of optical flow as a low-level task that relies on feature matching rather than high-level semantic understanding (e.g. the model can match similar textures). Consequently, optical flow models tend to generalize better to unseen domains.
>
> In practice, we did not observe any domain gap for the optical flow model. It appeared to perform well across datasets. It is also worth noting that both our method and SEA-RAFT were trained on many of the same datasets, including TartanAir and Spring, and very similar style datasets KITTI vs Waymo.

---

> > ### Author Response · Authors · 2024-11-20
> > **Official Comment by Authors (2/2)**
> >
> > **Q5: It would be beneficial if the authors provided depth evaluation results for static scenes and dynamic objects separately. This would clarify whether the performance improvement is due to coherent depth estimation on static scenes or the enhanced depth quality of dynamic objects.**
> >
> > Thank you for suggesting this analysis. As shown in the table below, MonST3R demonstrates significant improvements in depth estimation for dynamic objects, achieving **20.33%** and **41.78%** relative improvements in δ < 1.25 over CasualSAM and DUSt3R, respectively. The drastic performance gap between static and dynamic regions is consistent with the observations in **Table 1** of the CasualSAM paper. For DUSt3R, this performance difference is also explored qualitatively in **Figure 2 (right)** of our paper.
> >
> > It is worth noting that adding the flow loss, which is only applied to confident static regions, slightly reduces the depth accuracy of dynamic regions.
> > | **Method**           | **All**              |           | **Dynamic** |            | **Static**  |            |
> > |-----------------------|:-------------:|:-------------:|:-------------:|:-------------:|:-------------:|:-------------:|
> > |                       | Abs Rel ↓           | δ < 1.25 ↑ | Abs Rel ↓    | δ < 1.25 ↑ | Abs Rel ↓    | δ < 1.25 ↑ |
> > | CasualSAM             | 0.387               | 54.7      | 0.566        | 42.3       | 0.293        | 60.6       |
> > | DUSt3R                | 0.599               | 50.1      | 1.213        | 35.9       | 0.322        | 56.7       |
> > | MonST3R               | **0.335**           | **58.5**  | 0.478        | 50.9       | **0.270**    | **61.7**   |
> > | MonST3R (w/o flow)    | 0.339               | 57.7      | **0.468**    | **51.3**   | 0.277        | 60.9       |
> >
> > *Table 1: Video depth evaluation results on the Sintel dataset.*
> >
> >
> > ------
> >
> > **Q6: Do the authors use zero-shot models like Marigold and Depth-Anything? The use of the Waymo dataset (similar to KITTI) and the PointOdyssey dataset (with properties akin to Sintel) might explain why the proposed method outperforms these models. Can the authors specify which components of their approach are most critical for achieving this performance boost?**
> >
> > Thank you for your question. We do not use Marigold or Depth-Anything in our method. Regarding dataset distributions, we would like to clarify that the distribution of PointOdyssey (PO) differs significantly from Sintel. PO primarily contains indoor scenes with simple backgrounds (as shown on their [official webpage](https://pointodyssey.com/#dataset)), while Sintel mainly features outdoor scenes (21 out of 23 sequences) with more complex backgrounds. Additionally, for KITTI, we demonstrate that our method performs well even without fine-tuning on Waymo.
> >
> > | **Method**                         | **Abs Rel ↓** | **δ < 1.25 ↑** |
> > |------------------------------------|:-------------:|:--------------:|
> > | DUSt3R                             | 0.134         | 82.6           |
> > |                                    |               |                |
> > | MonST3R (PO+TA+Spring)             | 0.109         | 88.7           |
> > | MonST3R (PO+TA+Spring+Waymo)       | **0.104**     | **89.5**       |
> >
> > *Table 2: Video depth evaluation on KITTI dataset.*
> >
> > In terms of critical components, our ablation study (Tab. 5) highlights three main findings: (1) all training datasets contribute to improved camera pose and video depth estimation performance; (2) fine-tuning only the decoder and head achieves significantly better results compared to alternative strategies; and (3) the proposed optimization process noticeably enhance pose estimation with minimal impact on video depth accuracy. These factors collectively enable MonST3R to achieve its strong performance.

---

> > > ### Author Response · Authors · 2024-11-25
> > > **Follow-up of the Response**
> > >
> > > Thank you for the comments on our paper. We have provided a response and a revised paper on Openreview. Since the discussion phase ends on Nov 26, we would like to know whether we have addressed all the issues. If so, we would appreciate it if you would reconsider the score accordingly.
> > >
> > > Thank you.

---

> > > > ### Comment · Reviewer_VTp6 · 2024-11-25
> > > >
> > > > Thank you for addressing my concerns with additional experiments.
> > > > I increased my rating to 6 (positiver side )

---

> > > > > ### Author Response · Authors · 2024-11-26
> > > > >
> > > > > Thank you for your response and for updating your rating. Are there any specific concerns that you feel still place the paper on the borderline for acceptance? We would be happy to discuss or address any remaining issues to further clarify or strengthen our work.

---

### Official Review · Reviewer_CcMU · 2024-11-03

**Soundness:** 3
**Presentation:** 3
**Contribution:** 3
**Rating:** 6
**Confidence:** 4

**Summary:**

This paper presents MonST3R, a method for estimating geometry in dynamic scenes. It adapts the DUSt3R's pointmap representation to handle moving objects. Key points include: using suitable synthetic and real datasets for training despite data scarcity, introducing new optimization methods for downstream tasks like depth and pose estimation, demonstrating strong performance compared to prior works, and achieving good results in 4D reconstruction.

**Strengths:**

- MonST3R successfully adapts the DUSt3R's pointmap representation to dynamic scenes. By estimating a pointmap for each timestep and representing them in the same camera coordinate frame, it can handle the movement of objects.
- Data Efficiency: Trains well with limited data via strategies.
- Good Performance: Strong in video depth and pose estimation.

**Weaknesses:**

- There are also some high-quality synthetic datasets in the video field, such as DynamicStereo and IRSDataset. The author could consider expanding the scale of the training data to explore better model performance.
- There are doubts about the metrics used by the author when comparing video depth estimation. When the delta metric is below 0.7 or lower, it is usually considered that the prediction has completely collapsed. However, as far as the reviewer knows, the video depth estimation methods compared by the author have good performance and can provide basically correct depth estimation results at least on the Sintel dataset. Some metrics, such as NVDS, seem to be inconsistent with the report in the paper.
- It is necessary to explore the robustness of DUSt3R, that is, the failure cases when the scene is too dynamic.

**Questions:**

See weakness

---

> ### Author Response · Authors · 2024-11-20
>
> **Q1: There are also some high-quality synthetic datasets in the video field, such as DynamicStereo and IRS Dataset. The author could consider expanding the scale of the training data to explore better model performance.**
>
> Thank you for the thoughtful suggestion to consider additional datasets such as DynamicStereo and IRS. We appreciate the recommendation and agree that exploring diverse datasets is a valuable avenue for future work.
>
> Regarding these datasets:
>
> - **DynamicStereo**: While it shares a similar distribution to PointOdyssey (mostly indoor synthetic data), which we already utilize extensively, we will consider evaluating whether it adds meaningful diversity or performance improvements in future work.
> - **IRS Dataset**: This dataset focuses on static scenes and thus may have limited utility for training models aimed at dynamic scenes like MonST3R. Its coverage is also similar to TartanAir, which is already included in our dataset mixture.
>
> Given the current scope and storage constraints, we prioritized datasets that align most closely with the objectives of this work. Expanding the dataset scale and diversity to further improve model performance is an exciting direction in the future.
>
>
> ------
>
> **Q2: There are doubts about the metrics used by the author when comparing video depth estimation. When the delta metric is below 0.7 or lower, it is usually considered that the prediction has completely collapsed. However, as far as the reviewer knows, the video depth estimation methods compared by the author have good performance and can provide basically correct depth estimation results at least on the Sintel dataset. Some metrics, such as NVDS, seem to be inconsistent with the report in the paper.**
>
> Thank you for the valuable comment. We would like to clarify that our evaluation protocol closely follows that of DepthCrafter. The performance of the baseline methods in our results aligns with those reported in the DepthCrafter paper.
>
> It is important to note that we use **per-sequence alignment**, as opposed to the **per-image alignment** used by some other works, including NVDS (as evidenced in their [official implementation](https://github.com/RaymondWang987/NVDS/blob/main/vdw_test_metric.py#L195-#L244)). Per-sequence alignment is a more challenging evaluation setting, as discussed and demonstrated in the DepthCrafter paper. This difference in alignment protocol may explain why the reproduced NVDS results appear worse than the originally reported numbers.
>
> Below, we compare NVDS and MonST3R under both per-image and per-sequence alignment protocols on the Sintel dataset. As shown, per-sequence alignment is more challenging than per-image alignment, and MonST3R consistently outperforms NVDS under both settings.
>
> | **Alignment Method**    | **Method**         | **Abs Rel ↓** | **δ < 1.25 ↑** |
> |--------------------------|--------------------|:-------------:|:-------------:|
> | **Per-Image Alignment**  | NVDS (reported in [A])    | 0.335         | 59.1           |
> |                          | NVDS (reproduced)  | 0.329         | 59.5           |
> |                          | MonST3R           | **0.299**     | **61.2**       |
> |||||
> | **Per-Sequence Alignment** | NVDS (reproduced) | 0.408         | 48.3           |
> |                          | MonST3R           | **0.335**     | **58.5**       |
>
> *Table 1: Video depth evaluation on Sintel under different alignment protocols.*
>
> [A] *Neural Video Depth Stabilizer*, Wang et al., ICCV 2023.
>
>
> ------
>
> **Q3: It is necessary to explore the robustness of DUSt3R, that is, the failure cases when the scene is too dynamic.**
>
> We have included qualitative results highlighting DUSt3R's failure cases in dynamic scenes in **Figure 2** and **Figure A6 (rows 2-3)** of the original paper. Specifically, DUSt3R struggles to align image pairs correctly when the scene contains significant dynamic motion, as it often misaligns based on the foreground objects or puts the dynamic regions into the background. In contrast, MonST3R successfully aligns the images based on the static region.
>
> Quantitatively, the table below demonstrates DUSt3R's significant drop in performance on dynamic scenes, while MonST3R achieves a more robust performance.
>
> | **Method**           | **All**              |           | **Dynamic**       |           | **Static**        |           |
> |-----------------------|:-------------:|:-------------:|:-------------:|:-------------:|:-------------:|:-------------:|
> |                       | **Abs Rel ↓**       | **δ < 1.25 ↑** | **Abs Rel ↓**       | **δ < 1.25 ↑** | **Abs Rel ↓**       | **δ < 1.25 ↑** |
> | DUSt3R                | 0.599               | 50.1      | 1.213             | 35.9      | 0.322             | 56.7      |
> | MonST3R               | **0.335**           | **58.5**  | **0.478**         | **50.9**  | **0.270**         | **61.7**  |
>
> *Table 2: Video depth evaluation results on Sintel dataset.*

---

> > ### Author Response · Authors · 2024-11-25
> > **Follow-up of the Response**
> >
> > Thank you for the comments on our paper. We have provided a response and a revised paper on Openreview. Since the discussion phase ends on Nov 26, we would like to know whether we have addressed all the issues. If so, we would appreciate it if you would reconsider the score accordingly.
> >
> > Thank you.

---

> > > ### Comment · Reviewer_CcMU · 2024-11-29
> > >
> > > The authors addressed my concerns well, especially with regard to quantitative metrics, and I raise my score.

---

> > > > ### Author Response · Authors · 2024-11-29
> > > >
> > > > Thank you for your response and for raising your score! If you have any remaining concerns or questions that still place the acceptance at borderline, we would be happy to address them to further improve our work.

---

### Official Review · Reviewer_2c91 · 2024-11-03

**Soundness:** 4
**Presentation:** 3
**Contribution:** 3
**Rating:** 8
**Confidence:** 4

**Summary:**

This paper approaches the task of obtaining depth and camera pose given dynamic video. To do so, it begins from the DUSt3R architecture and pretraining. The paper argues optimizing dynamic scenes requires good estimation of foreground depth, and alignment based on dynamic objects; both of which DUSt3R struggles on due to static training data of mostly backgrounds. To this end, the paper fine-tunes on dynamic videos, as well as proposing slightly different optimization criteria. It shows state-of-the-art results on depth and camera pose estimation on several standard dynamic video benchmarks.

**Strengths:**

Clearly identifies and addresses key shortcomings of DUSt3R when applied to dynamic scenes
- This work identifies non-obvious weaknesses of DUSt3R in foreground depth prediction, in addition to more predictable poor dynamic correspondence performance. Experiments show clear improvement accordingly
- Getting MonST3R to work with dynamic points is nontrivial. Careful finetuning choices in data and network, along with regularizations to handle a point representation taking into account time must have been necessary to achieve such strong performance
- Comprehensive ablations clearly show impact of several contributions in data, training and inference. Notably, PointOdyssey (dynamic) is essential, and fine-tuning specifically the decoder and head is critical.

Effective and efficient performance on an important task
- SOTA performance on depth and camera pose estimation across standard dynamic video datasets: Sintel, TUM-Dynamics, Bonn, KITTI, NYU-v2.
- Impressive the method is still competitive on static scenes ScanNet, showing this fine-tuning does not sacrifice original DUSt3R results.

My rating of 8 (Accept) is because of strong experiment results (SOTA Depth and Pose), resulting from clearly identified weaknesses (static and distant training data), from a strong starting point (DUSt3R); weaknesses are minor.


***Post-discussion Update*** I leave this paper at an 8. The rebuttal well addresses weaknesses in this paper. I do not feel it is worthy of a 10 because the contributions are not exceptional, i.e. it consists mostly of fine-tuning on synthetic data for a sizeable but not massive gain; but results, writing and method are still very strong and the paper should be accepted.

**Weaknesses:**

(Minor) Camera Pose Estimation suite could be more comprehensive
- E.g. COLMAP or masked COLMAP baseline would be good benchmarks.
- Why are DROID-SLAM, DPVO and ParticleSfM not run on TUM, and DROID-SLAM and DPVO not on ScanNet?
- LEAP-VO is highly competitive with the proposed method. This is a minor weakness given this method also estimates depth and improves on DUSt3R, but should be considered when analyzing e.g. if this paper should be considered for strong accept.

**Questions:**

See Weaknesses

---

> ### Author Response · Authors · 2024-11-20
>
> **Q: Camera Pose Estimation suite could be more comprehensive**
> 1. **E.g. COLMAP or masked COLMAP baseline would be good benchmarks.**
> 2. **Why are DROID-SLAM, DPVO and ParticleSfM not run on TUM, and DROID-SLAM and DPVO not on ScanNet?**
> 3. **LEAP-VO is highly competitive with the proposed method. This is a minor weakness given this method also estimates depth and improves on DUSt3R, but should be considered when analyzing e.g. if this paper should be considered for strong accept.**
>
> Thank you for the detailed feedback. Below, we address each point raised:
>
> 1. **COLMAP and Masked COLMAP Benchmarks**
>
>    It has been reported in the ParticleSfM paper that (masked) COLMAP fails on 5/14 sequences on Sintel and 3/20 sequences on ScanNet. To provide a fair comparison, we evaluate MonST3R against (masked) COLMAP on the subset of sequences where COLMAP does not fail. As shown in the table below, MonST3R achieves superior performance across all pose evaluation metrics.
>
>    | **Method**            | **ATE ↓**       | **RPE trans ↓** | **RPE rot ↓**   |  | **ATE ↓**       | **RPE trans ↓** | **RPE rot ↓**   |
>    |------------------------|:-------------:|:-------------:|:-------------:|--|:-------------:|:-------------:|:-------------:|
>    |                        | **Sintel**      |                 |                 |  | **ScanNet (20 seq)** |                 |                 |
>    | COLMAP                | 0.145           | 0.035           | 0.550           |  | 0.143           | 0.064           | 1.384           |
>    | COLMAP+MAT [A]        | 0.069           | 0.024           | 0.726           |  | -               | -               | -               |
>    | COLMAP+Mask-RCNN [B]  | 0.109           | 0.039           | 0.605           |  | -               | -               | -               |
>    | **MonST3R**           | **0.041**       | **0.017**       | **0.312**       |  | **0.068**       | **0.018**       | **0.614**       |
>
>    *Table 1: Pose evaluation metrics on Sintel and ScanNet (subset of sequences where COLMAP does not fail). We do not compare with masked COLMAP in ScanNet because it is of static scenes.*
>
>    [A] *Motion-attentive transition for zero-shot video object segmentation*, Zhou et al., AAAI 2020.
>
>    [B] *Mask R-CNN*, He et al., ICCV 2017.
>
> 2. **Comparison with DROID-SLAM, DPVO, and ParticleSfM on TUM and ScanNet**
>
>    The LEAP-VO paper compares extensively against DROID-SLAM, DPVO, and ParticleSfM and shows that LEAP-VO outperforms these methods. We find that MonST3R, despite tackling a more difficult task, performs comparable to LEAP-VO, a state-of-the-art method for pose-only prediction.  We plan to include more comprehensive comparisons with these methods on TUM and ScanNet in future version.
>
> 3. **LEAP-VO Comparison and Intrinsic Sensitivity**
>
>    We would like to highlight that LEAP-VO requires ground-truth (GT) intrinsics as input. To analyze its sensitivity to intrinsics, we conducted experiments with approximated intrinsics, for example, focal length derived from image width, which is a common practice in the literature [C, D, E], or using the focal length predicted by MonST3R. The results show a noticeable performance drop for LEAP-VO when using noisy or estimated intrinsics.
>
>    | **Method**              | **ATE ↓**       | **RPE trans ↓** | **RPE rot ↓**   |  | **ATE ↓**       | **RPE trans ↓** | **RPE rot ↓**   |
>    |--------------------------|------------------|-----------------|-----------------|--|------------------|-----------------|-----------------|
>    |                          | **Sintel**      |                 |                 |  | **ScanNet**      |                 |                 |
>    | LEAP-VO w/ GT focal      | 0.089           | 0.066           | 1.250           |  | 0.070           | 0.018           | 0.535           |
>    |||||||||
>    | LEAP-VO w/ width focal   | 0.136           | 0.068           | 1.344           |  | 0.091           | 0.021           | 0.581           |
>    | LEAP-VO w/ MonST3R focal | 0.125           | 0.063           | 1.280           |  | 0.075           | 0.020           | 0.573           |
>    | **MonST3R**              | **0.108**       | **0.042**       | **0.732**       |  | **0.068**       | **0.017**       | **0.545**       |
>
>    *Table 2: Pose evaluation on Sintel and ScanNet datasets. Best results without using GT focal are **highlighted** .*
>
>
>    [C] *3D Photography using Context-aware Layered Depth Inpainting*, Shih et al., CVPR 2020.
>
>    [D] *3D Moments from Near-Duplicate Photos*, Wang et al., CVPR 2022.
>
>    [E] *SpatialTracker: Tracking Any 2D Pixels in 3D Space*, Xiao et al., CVPR 2024.

---

> > ### Comment · Reviewer_2c91 · 2024-11-22
> > **Rebuttal Response**
> >
> > Thanks to the authors for their additional comparisons. I think these can make the paper experiments more comprehensive. I feel the paper is still not worthy of a 10 for reasons in the updated "strengths"; but it should definitely be accepted (8).

---

> > > ### Author Response · Authors · 2024-11-24
> > >
> > > Thank you for your follow-up and recognition of the updates. We appreciate your constructive feedback!

---

### Official Review · Reviewer_GUwz · 2024-11-04

**Soundness:** 4
**Presentation:** 4
**Contribution:** 4
**Rating:** 8
**Confidence:** 5

**Summary:**

This paper extends DUSt3R, originally a feed-forward method for static scene reconstruction, to handle dynamic scenes by reconstructing scene geometry at each timestep. The initial performance of DUSt3R was poor due to its training on static datasets only. To improve results, this paper identifies and fine-tunes DUSt3R on several dynamic datasets. Consequently, the proposed method delivers competitive results in video depth estimation, camera pose estimation, and dense reconstruction. The approach is efficient and demonstrates results that rival those of specialized techniques.

**Strengths:**

Clarity and Readability: The paper is well-written, with a clear statement of the motivation, methodology, and challenges, making it easy to follow.

Effective Demonstration: The video demonstration is impressive, showcasing efficient 4D reconstruction with significant potential for downstream applications.

Sufficient Novelty: While the proposed approach builds on DUSt3R, extending it to dynamic scenes is non-trivial. The authors clearly outline the challenges and propose effective solutions to address them.

Convincing Evaluation: The evaluation is thorough, comparing the method to different categories of existing techniques, with both quantitative and qualitative results presented.

**Weaknesses:**

1. Mention of Related Work: For video depth estimation, self-supervised monocular depth methods such as Manydepth [a] and SC-DepthV3 [b] should be mentioned. SC-DepthV3, in particular, provides video depth estimation results on the Bonn dataset, which would make for a relevant comparison.

[a] The Temporal Opportunist: Self-Supervised Multi-Frame Monocular Depth, CVPR 2021.

[b] SC-DepthV3: Robust Self-Supervised Monocular Depth Estimation for Dynamic Scenes, TPAMI 2024.

2. Robustness in Dynamic Regions: The statement “For most scenes, where most pixels are static, random samples of points will place more emphasis on the static elements” is generally true, but there are edge cases that require more robust approaches. For example, in the Bonn dataset, moving objects such as humans can occupy over 50% of the image. Potential solutions could include masking out dynamic regions using segmentation masks or re-running relative pose estimation after identifying dynamic regions.

**Questions:**

See weakness

---

> ### Author Response · Authors · 2024-11-20
>
> **Q1: Mention of Related Work: For video depth estimation, self-supervised monocular depth methods such as Manydepth [a] and SC-DepthV3 [b] should be mentioned. SC-DepthV3, in particular, provides video depth estimation results on the Bonn dataset, which would make for a relevant comparison.**
>
> Thank you for the suggestion. We have discussed these papers in the revised Related Work section (Line 175).
>
> Additionally, we compared MonST3R with SC-DepthV3 on the Bonn dataset. As shown below, MonST3R outperforms SC-DepthV3:
>
> | **Method**   | **Abs Rel ↓** | **RMS ↓** | **δ < 1.25 ↑** | **δ < 1.25² ↑** | **δ < 1.25³ ↑** |
> |:-------------|:--------------:|:---------:|:--------------:|:---------------:|:---------------:|
> | SC-DepthV3   | 0.163         | 0.265     | 79.7           | 88.2            | 93.7            |
> | **MonST3R**  | **0.061**     | **0.209** | **96.6**       | **98.3**        | **99.1**        |
>
> *Table: Video depth evaluation on Bonn dataset.*
>
>
> ------
>
> **Q2: Robustness in Dynamic Regions: The statement “For most scenes, where most pixels are static, random samples of points will place more emphasis on the static elements” is generally true, but there are edge cases that require more robust approaches. For example, in the Bonn dataset, moving objects such as humans can occupy over 50% of the image. Potential solutions could include masking out dynamic regions using segmentation masks or re-running relative pose estimation after identifying dynamic regions.**
>
> Thank you for the insightful question. We have identified an issue in the explanation of our relative pose estimation method in the paper and have fixed it in the revision (Lines 251–260). Unlike traditional approaches, which rely on **cross-view correspondences** and thus dynamic regions will violate the static assumption, our method uses **same-view correspondences**, making it invariant to whether a region is static or dynamic.
>
> - **Relative Pose Estimation:** Traditional methods often rely on corresponding points across *two-views*, using techniques like the epipolar matrix or Procrustes alignment to recover the camera pose. These approaches are sensitive to dynamic regions as they violate the static assumption inherent to these methods. In contrast, our approach leverages the fact that the output pointmaps of the second image by MonST3R model is expressed in the camera coordinate system of the first frame. Since each point is directly aligned with its corresponding pixel, we can use the Perspective-of-N-Points method to recover the relative pose. To further improve robustness, we employ RANSAC and define a *valid mask* by thresholding the estimated confidence mask to exclude low-confidence points.
> - **Global Pose Optimization:** Our alignment loss $L_{\text{align}}$ (Eq. 3) leverages the correspondence between the global point cloud and the pairwise point cloud of the **same frame**, rather than using cross-frame correspondences. This design avoids the static assumptions required by cross-view correspondences, making the optimization process robust to dynamic regions.
>
> We appreciate your suggestion regarding the use of segmentation masks to improve robustness and agree that adding additional information like semantics might be an interesting and promising future direction for improving these systems.

---

> > ### Author Response · Authors · 2024-11-25
> > **Follow-up of the Response**
> >
> > Thank you for the comments on our paper. We have provided a response and a revised paper on Openreview. Since the discussion phase ends on Nov 26, we would like to know whether we have addressed all the issues. If so, we would appreciate it if you would reconsider the score accordingly.
> >
> > Thank you.

---

> > > ### Comment · Reviewer_GUwz · 2024-11-26
> > >
> > > Thanks for the update. It addresses my concerns. I think that the paper should be accepted and keep my original rating (8).

---

> > > > ### Author Response · Authors · 2024-11-29
> > > >
> > > > Thank you for your feedback and confirming our updates have addressed your concerns. We appreciate your support.

---

### Official Review · Reviewer_b6SW · 2024-11-04

**Soundness:** 3
**Presentation:** 3
**Contribution:** 3
**Rating:** 8
**Confidence:** 5

**Summary:**

The paper first identifies the limitations in joint pose and geometry estimation methods on dynamic scenes, and proposes to finetune based on the recent geometric learning model DUSt3R, representing the dynamic scenes with per-timestamp pointmaps. During global alignment for the pairwise inference results, trajectory smoothness loss and flow projection loss are added to the alignment term to adapt the optimization on sequential dynamic inputs. Experimental results show that the proposed method achieves state-of-the-art performance on the video depth estimation task and remains competitive in camera pose estimation, while jointly estimating both.

**Strengths:**

1. The method motivates the problem well on joint pose and geometry estimation on dynamic scenes by identifying the limitations in the current geometric learning method DUSt3R which is trained on static scenes.
2. With carefully selected datasets for dynamic scenes, together with a window-based training strategy, the scarcity of dynamic training data for finetuning is dealt with.
3. Finetuning results validate the proposed idea, with only the decoder and prediction head tuned, the model is capable of predicting pointmaps for the dynamic regions, showing sharping improvement compared to the DUSt3R with mask baseline.
4. The method achieves SoTA results on video depth estimation task, and competitive results on video pose estimation. The selected baselines are thorough, including baselines specialized in each task.

**Weaknesses:**

1. The loss used in fine-tuning is not detailed in the methods section. In line 294, "static mask is both a potential output and will be used in the later global pose optimization", the word "potential" here is confusing, is the static mask also used during finetuning?
2. Although a sliding window technique is used, the runtime reported is 30s for pairwise inference and 1min for global optimization given the 60 frames (around 2.5s) video input, which is slow.

**Questions:**

1. In the depth comparison in figure A1, the depth colormaps effectively hides the error, it's better to also add error colormaps. Additionally, in all the visualized frames, the model all predicts closer depth compared to ground truth on the dynamic parts. Is this specific to this dataset or is it a general bias of the model? What could be the possible reasons?
2. In lines 261-269, to calculate induced flow from camera motions, the method needs to estimate the intrinsics separately from the pointmaps. I wonder if using dataset intrinsics would degrade the performance?
3. Another related question on intrinsics, in lines 456-458, the relax of requiring intrinsics can be seen as a strength, however DUSt3R is not able to include known intrinsics. Is it considered to include the input intrinsics during the method development?
4. As we see from DUSt3R to MASt3R, the pointmaps serve as general and powerful representation for static geometry, as multiple downstream task can be estimated based on them. However, it's questionable whether it fit best for dynamic scenes. Here, an off-the-shelf flow estimator is needed to help obtain the static/dynamic mask, are flow fields themselves better suited for representing dynamics?

**Details Of Ethics Concerns:**

-

---

> ### Author Response · Authors · 2024-11-20
> **Official Comment by Authors (1/2)**
>
> **Q1: The loss used in fine-tuning is not detailed in the methods section. In line 294, "static mask is both a potential output and will be used in the later global pose optimization", the word "potential" here is confusing, is the static mask also used during finetuning?**
>
> We use the same confidence-aware regression loss as DUSt3R for fine-tuning the model. We have updated the paper to include this information in Line 244. The static mask is not used during finetuning, but it is one of the downstream applications derived from the pairwise pointmaps output from the model. The static mask is also used for the global alignment process, as mentioned in Line 338 and Eq. (5).
>
> ------
>
> **Q2: Although a sliding window technique is used, the runtime reported is 30s for pairwise inference and 1min for global optimization given the 60 frames (around 2.5s) video input, which is slow.**
>
> Compared to previous methods in the same category, MonST3R is significantly faster. For instance, CasualSAM requires approximately **60 minutes** and RCVD requires around **30 minutes** to process the same 60-frame sequence on the same hardware. Additionally, MonST3R's runtime is comparable to depth-only methods, e.g., DepthCrafter, which takes about 3 minutes to handle 60 frames of video.
>
>
> Currently, the majority of the run time is spent on global optimization and the intensive sampling of frame pairs. One interesting ablation is trying to eliminate these by running pointmap predictions between all the frames and an anchor frame. Then the global optimization would be unnecessary since all of the pairwise predictions would be in the camera coordinate frame of the anchor frame, and we can directly use the pairwise predictions as output. It also avoids extensive pair sampling since we only need to construct a single pair for each frame. Though this simplified approach has several limitations, such as sensitivity to the choice of the anchor frame and the lack of global information sharing across frames, it achieves **real-time reconstruction** at approximately **45 FPS** on a single RTX4090 GPU and offers a promising direction for fully feed-forward reconstruction from monocular video.
>
> We have included detailed explanations and additional examples in **Appendix B** of our revised version. Additionally, we have uploaded video examples of the real-time reconstruction results on our **[anonymous website](https://monst3r-paper.github.io/page0.html)**. While not included in our original submission due to its limitations, we believe this mode serves as an effective ablation study to highlight the efficiency of MonST3R.
>
>
> ------
>
> **Q3: In the depth comparison in figure A1, the depth colormaps effectively hides the error, it's better to also add error colormaps. Additionally, in all the visualized frames, the model all predicts closer depth compared to ground truth on the dynamic parts. Is this specific to this dataset or is it a general bias of the model? What could be the possible reasons?**
>
> Thank you for the valuable feedback. We have updated our revision to include **error maps** alongside the depth maps (Figure A.1) for better clarity and to address your concern regarding potential visualization bias.
>
> Regarding the observation that "the model predicts closer depth compared to ground truth on the dynamic parts," we believe this discrepancy arises from the visualization process. Specifically, min/max normalization was applied individually to each depth map in our previous version, and the presence of invalid pixels in the ground truth (GT) depth map can significantly bias the normalization, leading to an apparent discrepancy. In the revised version, we fixed it by using the same normalization scale for both the GT and all predicted depth maps. Based on the updated error maps and depth maps, it is clear that the foreground predictions are well-aligned with the ground truth.

---

> > ### Author Response · Authors · 2024-11-20
> > **Official Comment by Authors (2/2)**
> >
> > **Q4: In lines 261-269, to calculate induced flow from camera motions, the method needs to estimate the intrinsics separately from the pointmaps. I wonder if using dataset intrinsics would degrade the performance?
> > Another related question on intrinsics, in lines 456-458, the relax of requiring intrinsics can be seen as a strength, however DUSt3R is not able to include known intrinsics. Is it considered to include the input intrinsics during the method development?**
> >
> > Thank you for the insightful question. The DUSt3R framework theoretically allows the decomposition of its output point clouds into different intrinsics and poses, making it possible to specify intrinsics during optimization (as discussed in [DUSt3R issue #30](https://github.com/naver/dust3r/issues/30)). However, since the model does not take intrinsic parameters as input, the predicted point clouds are sometimes not fully faithful to the desired intrinsics. To address this, a potential direction for future work would be to develop an intrinsic-conditioned prediction model, which could enforce stronger alignment with specified intrinsics.
> >
> > To further explore this, we conducted experiments incorporating GT focal length information into the global optimization. The results, shown below, indicate that using GT focal improves performance on less challenging sequences but degrades performance on more challenging ones.
> >
> > | **Scene**            | **alley_2** | **ambush_4** | **ambush_5** | **ambush_6** | **cave_2** | |**avg**   |
> > |-----------------------|:-----------:|:------------:|:------------:|:------------:|:----------:|-|:---------:|
> > | MonST3R              | 0.0088      | 0.0511       | 0.0438       | **0.2128**   | **0.2651** |\| | **0.1163** |
> > | MonST3R w/ GT focal  | **0.0052**  | **0.0441**   | **0.0416**   | 0.2432       | 0.3501     |\|| 0.1368 |
> >
> > | **Scene**            | **scene707** | **scene708** | **scene709** | **scene710** | **scene712** || **avg**   |
> > |-----------------------|:------------:|:------------:|:------------:|:------------:|:------------:|-|:---------:|
> > | MonST3R              | 0.1057       | 0.0491       | 0.0545       | 0.0408       | 0.0543       |\|| 0.0609 |
> > | MonST3R w/ GT focal  | **0.0981**   | **0.0469**   | **0.0570**   | **0.0381**   | **0.0525**   |\|| **0.0585** |
> >
> > *Table: Camera pose estimation (ATE) evaluation on Sintel and ScanNet sequences.*
> >
> > ------
> >
> > **Q5: As we see from DUSt3R to MASt3R, the pointmaps serve as general and powerful representation for static geometry, as multiple downstream task can be estimated based on them. However, it's questionable whether it fit best for dynamic scenes. Here, an off-the-shelf flow estimator is needed to help obtain the static/dynamic mask, are flow fields themselves better suited for representing dynamics?**
> >
> > As discussed in L59-61, we take a geometry-first approach by estimating pointmaps because motion representations (e.g., 3D flow fields or trajectories) are typically harder to directly supervise due to the scarcity of annotated training data. In this work, we utilize an off-the-shelf flow estimator to obtain static/dynamic masks as a practical solution. However, we see a promising future direction in integrating motion as part of the model's outputs. Just as the DUSt3R authors expanded their method to include correspondences in MASt3R, we hope to similarly extend our approach to jointly predict geometry and dynamics in future work.
> >
> > In summary, while this paper emphasizes a geometry-first approach, we agree that jointly modeling geometry and motion is an exciting avenue for advancing 4D reconstruction.

---

> > > ### Author Response · Authors · 2024-11-25
> > > **Follow-up of the Response**
> > >
> > > Thank you for the comments on our paper. We have provided a response and a revised paper on Openreview. Since the discussion phase ends on Nov 26, we would like to know whether we have addressed all the issues. If so, we would appreciate it if you would reconsider the score accordingly.
> > >
> > > Thank you.

---

> > > > ### Comment · Reviewer_b6SW · 2024-11-27
> > > >
> > > > I thank the authors for the very detailed response and additional experiments.
> > > > The authors addressed my questions and comments very well.
> > > > Overall, the paper has a very solid contribution and compelling results that merit publication. I keep my rating to accept the paper.

---

> > > > > ### Author Response · Authors · 2024-11-29
> > > > >
> > > > > Thank you for your feedback and support! We are glad that our responses have addressed your concerns.

---

### Author Response · Authors · 2024-11-20
**General Response**

We thank the reviewers for their insightful comments, valuable feedback, and recognition of the strengths of our work:

- **Important Problem:**  Reviewers acknowledged the significance of the problem addressed in our work, highlighting its "significant potential for downstream applications" (GUwz) and describing it as “compelling and relevant” (VTp6).
- **Clear Motivation and Presentation:** Our paper was praised for "motivating the problem well" and "clear identification of limitations" in existing methods (b6SW, 2c91). Reviewers also highlighted its "clarity" and "easy to read" due to the "well-written" structure (GUwz).
- **Novel Adaptation of DUSt3R to Dynamic Scenes:** The "successful adaptation of DUSt3R's pointmap representation to dynamic scenes" and the “non-trivial extension” to handle moving objects were commended as a "reasonable and well-structured solution" (GUwz, 2c91, CcMU, VTp6). The adaptation was noted to involve "careful design choices" and "effective training strategies" to address the challenges of dynamic geometry estimation (b6SW, GUwz).
- **Strong Experimental Results and Efficiency:** Reviewers highlighted the "state-of-the-art performance" of our method on "depth and pose estimation tasks" and its "thorough evaluation" with comprehensive ablations (b6SW, GUwz, 2c91, CcMU, VTp6). The "impressive video demonstration" was also noted as a strength (GUwz). The paper’s "efficiency" in achieving results with "limited dynamic data" was commended (b6SW, CcMU).

------

We first summarize updates of the revised manuscript, followed by responses to individual comments. All revisions are highlighted in **red** in the updated version:

- **Related Work (GUwz-Q1; Line 175):** Added references for self-supervised video depth methods.
- **Fine-Tuning Objective (b6SW-Q1; Line 244):** Clarified the use of the same objective as DUSt3R.
- **Relative Pose Estimation (GUwz-Q2; Line 251-260):** Clarified the use of the PnP method.
- **Depth Map and Error Map (b6SW-Q3; Line 766-792):** Updated the error and depth maps for better visualization.
- **Real-Time Reconstruction (b6SW-Q2; Line 1026):** Provided detailed implementation and results (videos available on [anonymous website](https://monst3r-paper.github.io/page0.html)).
- **Details on Global Optimization (VTp6-Q1,2,3; Line 1080):** Elaborated on the details of $ L_{\text{smooth}} $ and $ L_{\text{flow}} $.

We look forward to further engaging discussions and appreciate the opportunity to refine our work based on the reviewers' constructive feedback.

---

### Meta-Review · Area_Chair_ujzS · 2024-12-14

**Metareview:**

The paper present an approach to depth and pose estimation from monocular video.
It extends DUST3R to handle dynamic scenes by fine-tuning on appropriate datasets and tailoring its global pose optimization to dynamic scenes. Specifically, a sliding-window based optimization improves run-time and losses encouraging smooth camera trajectories and consistency with optical flow improve accuracy.
The method compares favourably against several representative baselines for video depth and pose estimation. After clarifications provided in the rebuttal, all reviewers unanimously recommended to accept the paper. The AC agrees with this assessment.

**Additional Comments On Reviewer Discussion:**

Reviewers pointed out missing details on the fine-tuning loss, and requested clarification on the global optimization and reported results in comparison to prior work. All requests were met by the authors and reviewers raised their scores accordingly to unanimous accept.

---

### Decision · Program_Chairs · 2025-01-22

Accept (Spotlight)